

# The near wake development of a wind turbine operating in stalled conditions - Part I: Assessment of numerical models

Pascal Weihing[1], Marion Cormier[1], Thorsten Lutz[1], and Ewald Krämer[1]

[1]University of Stuttgart, Institute of Aerodynamics and Gas Dynamics, Pfaffenwaldring 21, 70569 Stuttgart, Germany

**Correspondence:** Pascal Weihing (pascal.weihing@iag.uni-stuttgart.de)

**Abstract.** This study comprehensively investigates the near wake development of a model wind turbine operating at a low tip speed ratio in stalled conditions. In the present paper, part I, different ways of representing the turbine, that is the full geometrical representation and the modeling by means of the actuator line method, but also different approaches for the modeling of turbulence are assessed. The simulation results are compared with PIV measurements from the MEXICO and NewMexico experiments. A highly-resolved numerical setup was created and a higher-order numerical scheme was applied targeting an optimal resolution of the tip vortex development and the wakes of the blades. Besides the classical unsteady Reynolds-averaged methodology, a recently developed variant of the Detached Eddy Simulation (DES) was employed, which features robust shielding capabilities of the boundary layers and enhanced transition to a fully developed LES state. An actuator line setup featuring the same turbulence modeling as for the DES simulation was created, where the aerodynamic forces were either evaluated by means of tabulated data or imposed from the averaged blade loads of the simulations with full blade geometry. The purpose is to distinguish between the effects of the force projection and the force calculation in the underlying blade-element method on the blade wake development. The averaged properties of the near wake flow field were accurately captured by all methods. Unsteady flow features of the separated shear layer were very well reproduced by the scale-resolving simulation with full geometric turbine representation. This method was also capable to predict the tip vortex development, in terms of vortex strength, position and size over the entire simulation domain. The classical URANS method performed very poorly. In the actuator line simulations it was found that very similar results are obtained compared to the fully resolved simulations, when inheriting their forces. However, when operating the actuator line with forces from blade-element momentum theory, the wake topology is not predicted correctly, in particular, in the inner part of the rotor. In the separated flow conditions of this case, it demands for correction models to take in to account the effect of rotational augmentation. In part II of the study the development and the dynamics of the early tip vortex formation is detailed in another paper.

## 1 Introduction

In the course of the total optimization of the levelized cost of energy, the turbines in the wind farms have to be placed ever closer to each other, so that distances in the order of 2-3 rotor diameters prevail in some wind directions. This means that the turbines are being placed more and more in the near wake, rather than the far wake. Compared to the far wake, which is mainly characterized by isotropic turbulence and where the influence of the detailed rotor aerodynamics plays a subordinate role, in



the near wake (1-2D) the effects of blade circulation and the transient events of the blade passage govern the unsteadiness in the wake. In particular, the tip and hub vortices characterize the flow topology of the wake. In addition, also viscous effects like flow separation can become relevant under certain operating conditions. In order to provide proof of the dynamic loads in such wake scenarios over the entire machine life time of 20 years, turbine manufacturers and park developers utilize engineering

wake models like Frandsen et al. (2006), Jensen (1983) and the dynamic wake meandering model Larsen et al. (2007). These models incorporate only a very simplified model representing the turbine and its operation. They are only valid in the far wake and exhibit inaccuracies in the near wake.

Therefore, it is essential to understand, improve, and further develop engineering models for the near wake. The starting point is to deepen the knowledge of the underlying physics and its simulative modeling. This is what this study aims to contribute

to. Computational Fluid Dynamics (CFD) shall be applied with different levels of model fidelity, in order to study the physical effects in the wake and to learn the limits of the respective method.

The test case at hand is the MEXICO model wind turbine operating in off-design conditions at low tip-speed ratio in which the rotor is exposed to stall. The unsteady flow phenomena involved there are particularly challenging to predict for flow simulation, in general. In order to assess different numerical models regarding the wake physics in such conditions,

two different methods shall be applied to represent the rotor. These are, first, the geometrically fully resolved approach and, second, the actuator line (ACL) method (Sorensen and Shen, 2002). The latter promises a significant computational time saving compared to the full-resolution simulation, since both the spatial and temporal resolution can be adapted to the chord of the blade instead of requiring the details of viscous boundary layers to be resolved. The method will be applied in two different modeling depths. First, in a "forced" mode, where the averaged forces from the full-resolution simulation are injected at the

rotating line. This is to examine to what extend the same circulation and wake properties can be achieved as with the full-resolution method, having exactly the same averaged aerodynamic forces at the blade in both cases. In a second step, the ACL is used classically with internal force calculation, with which the quality of the BEM for detached flows is to be verified. Regarding both modeling approaches, scale-resolving simulation techniques shall be applied and compared to classical RANS turbulence modeling, in order to expose the differences in the predicted wake topology.

## 1.1  Previous experimental and computational studies on the near wake

Experimental studies on the near wakes were mostly conducted in controlled conditions of wind tunnels. There, data acquisition by means of PIV or hot-wire anemometry is easy compared to velocity measurements in the field. Also, the inflow conditions are well known and controlled, both making this type of testing well suitable for the validation of numerical simulations. On the other hand, there is the drawback of scaling effects that results in relatively low Reynolds numbers and turbine designs that differ from modern multi mega-watt machines. Therefore, wind tunnel experiments based on small-scale or miniature

turbines (Chamorro and Porté-Agel, 2009, 2010; Bottasso et al., 2014; Wang et al., 2021) typically focus on the effects of the atmospheric boundary layer like shear, turbulence and stratification on the turbines' wakes. As discussed in Wang et al. (2021) the wakes of the scaled turbine with the real full scale machine agrees well in the far wake beyond four rotor diameters in terms





of wake deficit and turbulence statistics. The inconsistencies in the near-wake are caused by differences in swirl and circulation
distribution, as well as neglecting the effect of rotational augmentation.

Experiments with respect to the near wake of model wind turbines were conducted by numerous research groups (Grant and
Parkin, 2000; Ebert and Wood, 1999, 2001; Snel et al., 2007; Boorsma et al., 2014; Dobrev et al., 2008; Krogstad and Eriksen,
2013; Bartl and Sætran, 2017; Mühle et al., 2018; Micallef, 2012; Bastankhah and Porté-Agel, 2017; Okulov and Sørensen,
2007). The velocity deficits of the rotor blades a few chord lengths downstream of the blade was measured for example by
Ebert and Wood (2001); Schepers and Snel (2007); Micallef (2012). With that, a characterization of the induction effects of
the blade passage in the very near field of the blade could be performed.

The importance of tip vortices in the evolution and collapse of the wake has been further key focus for research in experiments
and numerical analyses. The vortex roll-up process directly around the blade tip surface was investigated by Micallef et al.
(2014). They found in PIV investigations on the Delft research turbine as well as by analyzing the data of the MEXICO
turbine (Schepers and Snel, 2007) that the tip vortex stays inward of the blade tip at early stages before any wake expansion
sets in. The vortex propagation in the rotor wake of the aforementioned MEXICO turbine was extensively studied within the
first MEXICO campaign in 2007 (Schepers and Snel, 2007). The tip vortices were tracked with the PIV technique on laser
sheets placed along the expected vortex trajectories in the wake for each operational point of the turbine. These ranged from
turbulent wake state over the optimum tip-speed ratio to a high wind speed case in the stalled regime, for both axial and yawed
inflow. In the MexNext projects (Schepers et al., 2012; Boorsma et al., 2018) the University of Stuttgart processed the PIV data
and extracted the vortex properties, such as position, strength (circulation) and size and compared different CFD solvers and a
free vortex code. In numerical studies using CFD, the tip vortices sizes were typically overpredicted compared to experiments.
In the results of Schulz et al. (2016), who used URANS simulations and a second order convection scheme (Jameson et al.,
1981) on a grid resolution of $\approx 16\,\%$ tip chord, the core radius was overestimated by a factor of five. In the simulations of
Nilsson et al. (2015), who employed the ACL method in an LES framework and a QUICK scheme on a grid resolution in the
tip region of $\approx 20\,\%$ tip chord, it was overestimated by approximately the same amount. Meister (2015) employed a vortex
refinement mesh together with a higher order WENO scheme (Jiang and Shu, 1996) with URANS. By that, the overprediction
could be reduced to a factor between three and four. In the studies of Carrión et al. (2014, 2015) the low-Mach number adapted
Roe-scheme (Rieper, 2011) was adopted together with URANS on wake resolution of $\approx 6.5\,\%$ tip chord, in order to reduce
dissipation of compressible schemes at low Mach numbers. In their studies the vortex core by a factor of around three. Other
parameters, such as vortex position, as well as integrated vortex circulation were accurately predicted by the aforementioned
studies. Here, the determination of the load distribution on the rotor blade is evidently sufficient to correctly reflect the thrust
and thus the wake induction and expansion as well as the circulation of the tailing tip vortices.

Further investigations on the tip vortex positions and the effect of vortex wandering were reported in Bailey and Tavoularis
(2008) and Dobrev et al. (2008). Those are important to understand the instability mechanisms that eventually lead to vortex
break down and mark the end of the near wake region. Using PIV measurements and conditional averaging they separated the
mean field from the vortex wandering and were able to map the variance of the vortex position. The vortex wandering was
also evaluated in the study of Sherry et al. (2013), who investigated the wake of the Tjaeborg wind turbine in a water tunnel



at different tip-speed ratios. They found higher standard deviations of the vortex wandering for lower tip-speed ratios. The

phenomenon of vortex wandering was recently also extracted for the larger BeRT turbine of TU Berlin by Soto-Valle et al. (2022). Theoretical studies on the instability mechanisms leading vortex breakdown were conducted by Ivanell et al. (2010); Sarmast et al. (2014); Kleine et al. (2022) using ACL simulations and controlled perturbations. By means of modal analyses the authors found the amplification of waves traveling along the vortex spiral that trigger the mutual inductance instability of the tip vortices.

The formation and development of the root vortex was investigated in the experimental study of Akay (2016) with PIV measurements, from which a three-dimensional reconstruction of the flow field was derived (Micallef, 2012; Akay, 2016). The study of Akay (2016) investigated the velocity and vorticity field with special attention to the root vortex and related it to the spanwise flow around the blade. This test case was numerically studied by Herráez et al. (2016). In their study using the RANS approach they investigated the formation of the root vortex in the very inner blade sections by comparing the velocity fields

close to the blade surface with measured PIV data. They found overall good agreement on the velocity components already $10\,\mathrm{mm}$ above the blade surface, suggesting that for design conditions the RANS approach correctly captures the vortex roll-up mechanism.

Wind tunnel experiments for validating aerodynamic simulation tools were conducted within the "Blind Test" campaigns that were organized at NTNU (Krogstad and Eriksen, 2013; Bartl and Sætran, 2017; Mühle et al., 2018). In these experiments

the complexity was increased from campaign to campaign, starting with a single wind turbine at uniform inflow, adding wind shear and turbulence, up to yawed conditions and wake interaction. In the comparison of the numerical models with measurements, it turned out that a full-turbine representation in conjunction with Detached Eddy Simulations produced superior results especially in highly complex flow situations such as yawed and partial wake situations with respect to the velocity and turbulence field compared to RANS and actuator disc approaches. Actuator line simulations presented in Mühle et al. (2018)

using experimental airfoil data and high grid resolution in combination with LES performed similarly well as the geometrically fully resolved IDDES simulation.

The experimental results of the MEXICO and NewMexico experiments were utilized as validation for a large number of numerical studies and collected in Boorsma et al. (2018). The computational approaches ranged from lifting-line codes (Grasso et al., 2011), actuator disc and actuator-line methods (Sarmast et al., 2016; Nilsson et al., 2015) to full-turbine representations,

which were simulated with the RANS (Lutz et al., 2011; Meister, 2015; Schulz et al., 2016; Bechmann et al., 2011; Sørensen et al., 2016; Carrión et al., 2014), hybrid RANS/LES methods (Weihing et al., 2016) and more recently also with the Lattice Boltzmann method (Li et al., 2021). As summarized by Boorsma et al. (2018), CFD simulations led to superior results in complex situations such as in stalled conditions, however, also to larger scattering among the different contributions. This implies that careful selection of the numerical input parameters, models and grid resolution is essential to achieve accurate

predictions for the simulation problem at hand.

In order to evaluate the way of the turbine modeling approach within a single numerical framework, Troldborg et al. (2015) compared the near wake of actuator disc and line simulations with blade resolved simulations for the NREL 5M reference turbine in uniform and non-sheared turbulent inflow using both, the RANS and the DES method. In the near wake, up to



one diameter downstream of rotor, all three approaches resulted in very similar mean velocity profiles. However, the velocity fluctuations induced by the tip vortices were predicted lower in the ACL simulations and were by nature absent in the actuator disc results. In the transition from near wake to far wake the breakdown of the tip vortices was predicted far too late in the actuator-type simulations. In turbulent inflow, the large scale disturbances imposed by the ambient turbulence led to very comparable breakdown mechanism of the wake in all three simulations and accordingly to very similar velocity and turbulence profiles. Although in the near wake at one rotor diameter into wake, the actuator line method had the tendency of predicting higher velocity fluctuations in the hub region, but larger fluctuations in tip region compared to the fully resolved simulation. Good agreement of actuator-line and fully resolved simulations under turbulent inflow was also found in the study Weihing et al. (2017), where the ACL was compared to fully resolved simulations at high turbulence ambient inflow in complex terrain and a lower turbulence environment representative for offshore conditions. In both cases very similar wake topologies and breakdown phenomena could be observed, which lead to good agreement in the velocity and turbulence profiles, already in the near wake region. More recently Hodgson et al. (2022) compared ACL and blade resolved simulations with the measurements gained in the DanAero experiment (Bak et al., 2010) for the NM80 turbine for axial and yawed inflow conditions. They employed the ACL coupled to the solver Flex5 in order to capture also aeroelastic effects like blade torsion and deflections. In the outer blade region the loads were predicted lower than in the fully resolved simulations, but closer to the experimental data. The wake pattern and profiles were predicted similarly in the turbulent inflow cases for both approaches, despite the ACL was run on a coarser grid which dissipated the discrete tip vortices rather quickly.

## 1.2 Objectives and Outline

The following objectives shall be addressed within part I of the paper:

1. The benefit of scale-resolving methods for prediction of the near wake flow field shall be investigated for stalled conditions.

2. It shall be examined, whether the ACL method can be applied to stalled operation and how accurate the predictions are with respect to fully resolved simulations and PIV experiments.

3. The ACL line with forces injected from the fully-resolved simulations shall be compared, to evaluate whether the same near blade velocity and circulation distribution is obtained.

4. It is targeted to predict the tip vortex evolution as close as possible to reality by employing scale-resolving simulations on a highly-resolved numerical setup.

In part II of the study, the formation process of the tip vortex and the development of secondary vortex structures will be investigate in detail.

The next section describes the experimental test case of the MEXICO turbine and the operation conditions. This is followed by information on the flow solver and the numerical modeling in section 3 and on the employed computational setups in





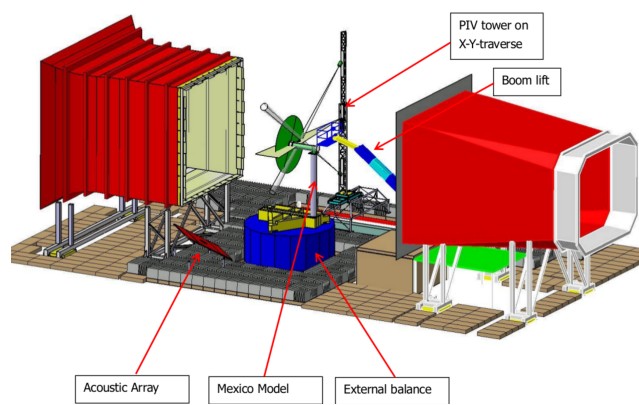

**Figure 1.** Sketch of the MEXICO test setup in the LLF (DNW) (Philipsen et al., 2014) (with courtesy from Schepers).

section 4. The results are presented in section 5, starting with the wake in the vicinity of the blade, moving to the rotor wake and finally discussing the evolution of the tip vortices. The main conclusions are summarized in section 6.

## 2 Test case

The MEXICO model wind turbine was experimentally studied in Large-Low-Speed Facility (LLF) of the Deutsch Nieder-ländische Windkanäle (German Dutch Wind Tunnels, DNW) within two test campaigns. The first campaign took place in 2006 (Snel et al., 2007) and shall be referred as "'MEXICO" campaign. In 2014 a second campaign was conducted with the same wind tunnel model and which is called "New Mexico" experiment (Boorsma and Schepers, 2015).

In principal, the basic test setup was the same in both campaigns. The arrangement is sketched in Fig. 1. The tunnel was operated in an open-jet mode with a contractor equipped with Seifert Flügel at the lip and a collector, resulting in a cross section of $9.5 \times 9.5\,\text{m}^2$. The MEXICO model was mounted on a test rig, connected to the floor of the hall by the external balance. The PIV system was mounted on a tower located outside of the cross section and which enabled traversing in the horizontal plane. The camera arrangement was stereoscopic, in order to obtain three velocity components in a two dimensional slice in the $x$-$y$ plane.

The MEXICO model turbine comprises three blades of a radius of $2.25\,\text{m}$, the tower and the nacelle. The latter includes the hub which is a rotating ring located downstream of the non-rotating "spinner" and houses the generator in the rear part. The tower is located $2.13\,\text{m}$ downstream of the rotor plane and has a diameter of $0.51\,\text{m}$. The blades of the model are instrumented with Kulite pressure sensors distributed over the five radial sections $r/R = 0.25; 0.35; 0.60; 0.82; 0.92$.

The blade is composed of three different airfoils. The DU91-w2-250 is installed in the root region ($0.2 < r/R < 0.44$), the Risø-A1-21 in the mid region ($0.55 < r/R < 0.65$) and the NACA 64-418 in the outboard section ($r/R > 0.74$). The shape of the blade tip is an ellipsoid. In the transition zones, the shape of the blade is linearly blended between the different airfoils. The chord and twist distributions, as well as the sectional Reynolds and Mach numbers are shown in Fig. 2. The maximum



**Table 1.** Operational conditions for the MEXICO test cases in axial inflow.

| Parameter | Value |
|---|---|
| Wind speed [m/s] | 24.05 |
| Temperature [K] | 294.25 |
| Density [kg/m$^3$] | 1.195 |
| Rotor speed [rpm] | 425.1 |
| TSR $\Lambda$ | 4.2 |
| Pitch angle $\beta$ [°] | $-2.3°$ |
| Forced transition | $r/R < 0.7$ |
| Natural transition | $r/R > 0.7$ |

tip speed of around $100\,\mathrm{m/s}$ and the chord distribution were elaborated to maximize the Reynolds number, while keeping the Mach number reasonably low. The Reynolds number is relatively constant over the radius and varies between $0.48$ and $0.64e^6$. The Mach number, which is estimated from the kinematic inflow velocity, increases linearly and reaches a maximum value of around $0.3$ at the blade tip. Compared to modern full-scale turbines the Reynolds numbers are still more than one order of magnitude lower, whereas the Mach number is already slightly above. With lower Reynolds number the area of separation and its resolved structures are expected to be relatively larger compared to full-scale turbines. Additionally, a clear effect of compressibility can be expected, namely an increasing adverse pressure gradient and therefore an earlier separation than at full scale.

In contrast to the former MEXICO experiments (Schepers and Snel, 2007), where zig-zag tapes were applied over the entire blade radius, in the New MEXICO experiments tripping was only applied to the inner rotor sections ($r/R < 0.7$) (Boorsma and Schepers, 2015). For the outer rotor portion ($r/R > 0.7$) clean conditions allow for natural laminar to turbulent transition of the boundary layer.

### 2.1 Operational conditions of the reference cases

The reference case for the simulations presented is an off-design case at low tip-speed ratio. The pitch settings cause flow separation along the entire blade radius. The operational point is taken from Boorsma and Schepers (2017) and summarized in Tab. 1.

It could be argued that such an off-design case has no practical relevance, since the turbine is supposed to operate outside of any stall issues. However, it is precisely the challenging flow patterns, where CFD is applied in industry and where it makes a difference, compared to engineering approaches. It is therefore highly important to thoroughly assess the accuracy and also the best-practices of CFD, especially with regard to evolving methods such as scale-resolving simulation techniques. This test case offers an excellent data basis with which the wake behavior and in particular the blade tip vortices can be analyzed in detail.





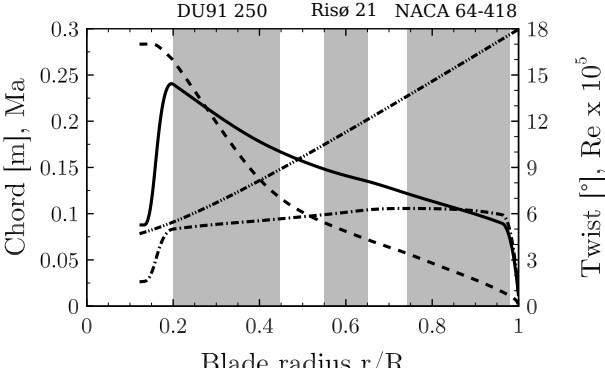

**Figure 2.** Radial distributions of the airfoil chord (——), twist angle (- - -), chord Reynolds (-·-·-) and inflow Mach number (-··-··-). The grey patches indicate the locations of the different airfoil types.

## 3 Flow solver and turbulence modeling

The simulations were performed with the block-structured, compressible flow solver FLOWer. The code was developed by the German Aerospace Center (DLR) (Kroll et al., 2000) and has undergone continuous development at IAG facilitating the understanding of multidisciplinary flow phenomena of wind turbines. Relevant extensions for the present study are the higher-order numerical methods (Kowarsch et al., 2013), scale-resolving simulation techniques (Weihing et al., 2016, 2020), and the actuator line method (Weihing et al., 2017; Bühler et al., 2018; Cormier et al., 2021).

Two turbulence modeling approaches are investigated within the present simulations. The first is the unsteady RANS method based on the SST turbulence model (Menter, 1994). The location of laminar to turbulent transition is predicted using the amplification factor model of Coder (2014), which is coupled to the turbulence model by a transport equation of the intermittency variable similarly as conducted by Menter et al. (2015). The amplification model is based on elements of linear stability theory (Drela and Giles, 1987) and requires the specification of a critical amplification factor $N_{crit}$, which is set to 9 in the present study. Note that the transition model was zonally deactivated in those regions that are affected by the zig-zag tape. Thus, it is deactivated in the radial region $r/R < 0.7$, where we have a zig-zag tape, but only downstream of it in the chordwise direction (10 % chord on suction and pressure side). The second approach is a recently developed variant (Weihing et al., 2020) of the Detached Eddy Simulation (DES). The main goal of DES is to treat the boundary layers in the RANS mode while resolving the separated flow structures in a LES-like behavior. In DES97 (Spalart, 1997) and also in Delayed DES (Spalart et al., 2006), it was observed that the LES domain may invade the boundary layer once the grid is refined. When this happens, the modeled stresses are reduced. Eventually, this leads to so-called grid-induced separation in flows with adverse pressure gradients. In the recently developed variant (Weihing et al., 2020) used in the present study, the interface between RANS and LES domains is estimated based on a localized version of the Bernoulli equation. Therefore, the method is called BDES. It identifies the presence of resolved turbulence using a shear layer detector and the $\sigma$ velocity gradient operator of Nicoud et al. (2011). This shielding method has proven to be very robust against grid-induced separation effects even under asymptotic grid refinement





and adverse pressure gradients and has been successfully implemented and applied to separated flows in other solvers (Fuchs
and Mockett, 2020; Ehrle et al., 2022). Furthermore, the velocity gradient operator within the turbulence model is replaced by
$\sigma$-operator, once it is in LES mode (Mockett et al., 2015). Together with a filter width $\Delta_{\tilde{\omega}}$, this reduces the eddy viscosity in
areas of pure shear and solid rotation and thereby yields a faster development of the hybrid model towards a mature LES flow
field. It should be noted that the LES model is only activated once the transition model is effectively in turbulent mode, in order
to prevent an unconstrained balance of production and destruction terms in the turbulence model.

Regarding the convection schemes, the inviscid fluxes are approximated by the higher-order WENO scheme of Martín
et al. (2006), implemented with four-point stencils. The scheme features its optimal weights such that maximum bandwidth
resolution is achieved rather than pure minimization of the truncation error. The viscous fluxes were approximated by central
differences. Time integration is performed by explicit multi-stage Runge-Kutta schemes. A second order implicit dual-time
stepping algorithm according to Jameson (1991) is employed for time accurate computations.

Moreover, the ACL approach (Sorensen and Shen, 2002; Mikkelsen et al., 2003) is used to include the effects of the blade
aerodynamic forces as momentum sources in the Navier-Stokes equations. The two dimensional aerodynamic forces at each
blade element are distributed over the neighboring computational cells using a Gaussian function. In the current implemen-
tation, the loads can be either computed from the local flow field properties and tabulated airfoil data, in alignment with the
original formulation, or forced as a steady blade loading from an external radial force distribution. Further details on the ACL
are found in 4.2.

## 4 Computational setup

Based on the experience gained during the MexNext projects a computational setup based on a one-third model was created.
The impact of the tower, which is located almost one rotor radius downstream was considered as subordinate regarding the loads
and near wake behavior in uniform conditions. The numerical representation of the wind-tunnel geometry was not considered,
as the study of Réthoré et al. (2011) showed negligible influence on the velocity field upstream of the rotor, which can be
therefore assumed uniform. Also, the wake development is not substantially affected by the tunnel environment.

### 4.1 Grids

The grid used to resolve the flow field in the vicinity of the rotor blade consists of four parts which are sketched in Fig. 3:
The near-body grid covers the attached and separated boundary layers and extends around one quarter chord from the blade
surface. It is embedded into a Cartesian background grid with hanging nodes that refines the separated wake region and rotates
with the blade. A tip cap closes the blade mesh and is required for topological reasons. At the root, a blade-connector grid is
used to enable pitching of the cylindrical root relative to the nacelle. The near-body grid is constructed in an O-type topology,
that was created by hyperbolic extrusion. The grid spacings, which are summarized in Table 2 were specified according to the
recommendations of AIAA's drag prediction workshop guidelines for aircraft wings (Vassberg et al., 2008). In order to keep
the number of cells manageable, the amount of grid points was higher on the suction side (321 nodes) than on the pressure side



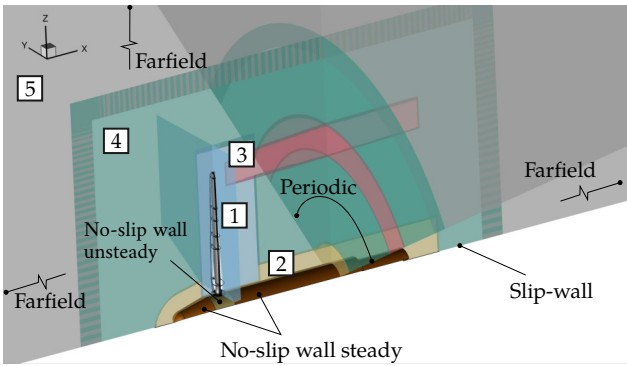

**Figure 3.** Overlapping grid structures and boundary conditions of the simulation setup for the MEXICO one-third model wind turbine.

**Table 2.** Cell spacings and grid information

|  | Fully resolved | ACL |
| --- | --- | --- |
| Turbine representation | Resolved geometry | Momentum sources |
| Blade mesh topology | O-mesh | Not required |
| Spacing at leading edge | $0.1\%c_{local}$ | – |
| Spacing at trailing edge | 2 %TE thickness | – |
| Maximum $y^+$ value | $< 1$ | $< 1$ @nacelle grid |
| Chordwise points, suction side | 321 | – |
| Chordwise points, pressure side | 161 | – |
| Radial points | 1377 | – |
| Normal points | 113 | – |
| Spacing, tip-vortex area (ax, rad, tan) | $0.004 \times 0.005 \times 0.006\,\mathrm{m}^3$ | |
| Spacing, rotor wake | $0.01\,\mathrm{m}$ | |
| Grid cells: tip vortex refinement (M) | 47 | |
| Grid cells: blade (M) | 80 near-body | – |
|  | + 220 refinement | 10 refinement |
| Grid cells: nacelle (M) | 18 | |
| Grid cells: background (M) | 40 | |
| Grid cells in total (M) | 424 | 118 |

(161 nodes). However, this required the construction of a cap mesh at the blade tip to close the grid. In the radial direction, the grid spacing is adapted to $1\%$ local chord. The near-body grid is embedded into a Cartesian background mesh which rotates with the blade and serves as a refinement of the separated airfoil wake. It is extended around $1.5$ local chord lengths downstream and the spacing is around $0.7\%c_{mean}$.





The grid of the nacelle, labeled with $\boxed{2}$ in Fig. 3, consists of a one-third periodic surface segment, which however does not use a singular line, due to problems of that boundary condition with the WENO scheme. Instead, a very small angular segment is created near the axial center that is treated as a slip-wall. The extrusion covers the region of the root vortex and the spacing is adjusted to match the resolution in the background mesh.

The tip vortex refinement grid, labeled with $\boxed{3}$ in Fig. 3, was designed based on the findings made in the MexNext III project Boorsma et al. (2018) in the IAG work package "Near Wake Aerodynamics". The centerline of the present tip vortex mesh was therefore placed along the core positions evaluated in those simulations. Since the vortex core radius was overpredicted in MexNext III in own simulations by around a factor of two, even for the numerical schemes involving the smallest numerical dissipation, the resolution was doubled in the present study, in order to examine whether more accurate predictions can be obtained. The specified cell size is $0.004 \times 0.005\,\mathrm{m}^2$ normal to its axis and averages a cell size of $0.006\,\mathrm{m}$ in the circumferential direction. The largest cell size is therefore around $10.6\,\%$ of the chord in the tip region. The grid extends $1.5\,\mathrm{R}$ in the axial direction, which requires $689 \times 97 \times 721$ points in the axial, radial and circumferential direction.

Two background meshes have been created to fill the computational domain towards their farfield boundaries. The near-background grid, labeled with $\boxed{4}$ in Fig. 3 covers the close vicinity of the rotor. It extends around one rotor radius upstream and 2.2 radii downstream and maintains a quasi-constant grid spacing of $0.01\,\mathrm{m}$ which is around $6.6\,\%$ of the mean chord. The far-background mesh adopts the resolution of $0.1\,\mathrm{m}$ in the overlapping region of the near grid and coarsens the mesh towards the farfield. The latter is placed $18\,\mathrm{R}$ upstream and $22\,\mathrm{R}$ downstream of the rotor and at a distance of $18\,\mathrm{R}$ in the radial direction. In order to omit the singular line in the axial center both background grids use an inviscid surface patch placed in a very small distance from the actual center line.

For the actuator line (ACL) setup, the geometry-fitting grid of the nacelle, the background meshes and the tip vortex refinement mesh, labeled with $\boxed{2}$, $\boxed{3}$, $\boxed{4}$ and $\boxed{5}$ in Fig. 3, are used. Additionally, a Cartesian local refinement mesh with a cell size of $0.004 \times 0.005 \times 0.006\,\mathrm{m}^3$ in the actuator line area is utilized. This mesh is extruded towards the boundaries to match the cell size of the resolution of the near-background grid in the Chimera overlap region.

The relevant grid spacings and cell numbers are summarized for the individual simulation setups in Tab. 2.

## 4.2 Actuator line method

The ACL setup aims at determining the limits of the actuator line approach to simulate the flow physics in the rotor wake at the present low-induction operating conditions. To assess the effects of the ACL modeling solely, only the blade is replaced by an actuator line. The nacelle boundary layer is resolved using the BDES scheme, as in the fully resolved setup, referred as FR-BDES in the following.

In its original formulation, the ACL approach can be broken down into two steps. First, in step i), the blade loading is computed at each time step based on the local velocity vector - extracted from the CFD flow field at the blade position - and tabulated airfoil data. Then, in step ii), the corresponding volume forces are smeared over the computational cells around each blade element via a Gaussian distribution function, to avoid numerical instabilities.



In order to address objective 2 formulated in Sec. 1.2, in the present study, the effects of these two steps are separately investigated. First, solely the effects of step ii), i.e. the force projection on the computational setup, are evaluated. The time-averaged blade loads as resulting from the blade-resolving FR-BDES simulations are imposed as steady blade loads. This case is referred as ACL-forced. Then step i), i.e. the blade loads computation from the local flow field and airfoil data, is activated. In this case, both effects of loads computation and force distribution are considered, as in the commonly applied ACL approach. This case is referred in the following as ACL-polars or standard ACL.

Along the blade radius, 125 actuator nodes are employed. In order to better capture the gradients towards the blade tip and blade root, the node spacing is reduced towards the inner and outer blade extremities. A 3D isotropic Gaussian projection function is applied, which projection width $\epsilon$ is proportional to the local airfoil length. According to the 2D investigation of the optimal projection width by Martínez-Tossas et al. (2017), a factor of $0.23$ is used, $\epsilon = 0.23 \times c$. Moreover, in order to avoid numerical instabilities, it is ensured that the Gaussian width $\epsilon$ is larger than two computational cells at any point, $\epsilon \geq 2\Delta x_{min}$.

For the ACL-forced-polars case, the experimental airfoil data from the MEXICO project are used (Schepers and Snel, 2007). The local velocity vector is extracted from the CFD flow field using 8 sampling points equidistantly located on a circular line around each ACL node, as suggested in Bühler et al. (2018). This approach is based on the LineAve method (Jost et al., 2018), originally developed to extract the local angle of attack from fully resolved CFD simulations. It has been shown that using this approach together with a denser nodes distribution towards the blade tip, the loads are correctly predicted in the blade tip area Bühler et al. (2018). Thus, no blade tip correction model is applied. Due to the investigated off-design rotor operating conditions, the Beddoes-Leishman dynamic stall model is applied (Leishman and Beddoes, 1989).

### 4.3 Boundary conditions and numerical settings

The boundary conditions are also represented in Fig. 3. The $120°$ domains are mapped periodically and the global domain boundaries carry farfield conditions. As already mentioned the small patch near the axial center is treated as inviscid wall. All other surfaces have no-slip conditions. On the nacelle, only the middle beige part is rotatable, while the rear and front parts (brown colored patches) remain fixed. This is achieved in the rotating grid setup without Chimera or sliding mesh interfaces. For this purpose, the rotational velocity of the surface boundary is subtracted at the first interior cell. Then, the negative value of the interior cell is set on the ghost layers as for a standard no-slip wall. This boundary condition was already set in the one-third MEXICO simulations of Meister (2015). However, in those simulations the entire front portion was rotating.

The time step was set to $\hat{\approx} 1/32°$, in order to resolve one convection over the blade at $r/R = 0.9$ by approximately 90 time steps. Note that these parameters refer to the production runs of the simulations. During the initial phase, steady state simulations, larger time steps and higher dissipative schemes have been employed, in order to accelerate convergence and to enhance numerical stability.

The simulations of the fully resolved turbine were performed on 12960 *Intel Xeon* E5-2680 v3 cores of the Cray XC40 (Hazel Hen), whereas the ACL was performed on *AMD EPIC* 7742 cores of the Hewlett Packard Enterprise Apollo (Hawk) machine, both at the High-Performance Computing Center Stuttgart. Data are extracted for one rotor revolution which corresponds to 11520 time steps.



# 5 Results

## 5.1 Overview

In this section a general overview on the wake formation will be given, with view on the overall topology as well as on the
330 tip and root vortices, discussing qualitatively the flow patterns predicted by the individual simulation models. Following that,
the blade loads and the circulation will be evaluated, since these are important quantities to understand the evolving wake
phenomena. The wake is then analyzed first in the surrounding of the blade and then on the scale of the rotor by comparing the
predicted flow fields with measurements from PIV. Special focus is on the development of the tip vortices.

### 5.1.1 The structure of the wake

The general wake topology of the rotor is shown in Fig. 4 by a comparative vortex visualization of the different simulations
and the experiment. In the simulations, vortices are visualized by iso-surfaces of $\lambda_2$, whereas in the experiment, smoke was
injected near the tip of two of the rotor blades (Boorsma and Schepers, 2014). Therefore, in the experiment, not every helix
ring can be compared with the simulations. First of all, the small expansion of the wake and the large pitch of the tip vortex
helix can be observed, which are a direct consequence of the low tip-speed ratio and induction of the rotor. On a real turbine
operating, for example in offshore conditions with low turbulence intensity, such a state would result in very long wakes, since
a breakdown due to mutual tip vortex interaction becomes very unlikely. Apart from that, the experiment indicates that the
tip vortex consists of two regions: there is a very narrow area, in which the density of the smoke is high and which can be
associated with the vortex core. This core region is surrounded by less dense, frayed out smoke, a "turbulent halo", which is an
indicator of secondary vortex structures developing around the main tip vortex.

With the fully resolved turbine geometry and BDES turbulence modeling, the most detailed picture of the evolving flow
structures of the blade wake and tip vortex region is given. This impressively exemplifies the multi-scale problem of turbulent
flow around wind turbines. Very small scale structures appear in the vicinity of the blade and emerge from the Kelvin-Helmholtz
instability of the separated boundary layer. The latter is triggered by the off-design operational point of the rotor. Downstream
of the rotor blade, the unstable shear layers from the suction side and the trailing edge interact and break the structures into
350 three-dimensional turbulence. The main shedding mode of the blade wake is associated with the van Kármán instability. The
sizes of the vortex structures are largest towards the root and decrease in size towards the tip, since there, chord decreases
and the outer velocity increases. Both result in smaller integral length and time scales. Longitudinal structures of higher
velocity indicate trailing vortices at different radial positions, which, however, are strongly superimposed by turbulence. Those
longitudinal structures are indicative of spanwise gradients of bound circulation. Similarly to the experiments, the tip vortices
reveal a compact vortex core, which is surrounded by secondary structures. Chaderjian (2012), who found them in rotorcraft
simulations as well, gave them the nickname "turbulent worms". Here they appear to arise from the turbulent shear layer that
wraps around the main vortex. The direction of the "worms" is perpendicular to the axis of the main helix. A more detailed
analysis on the formation of these structures will be elaborated in part II of the study.



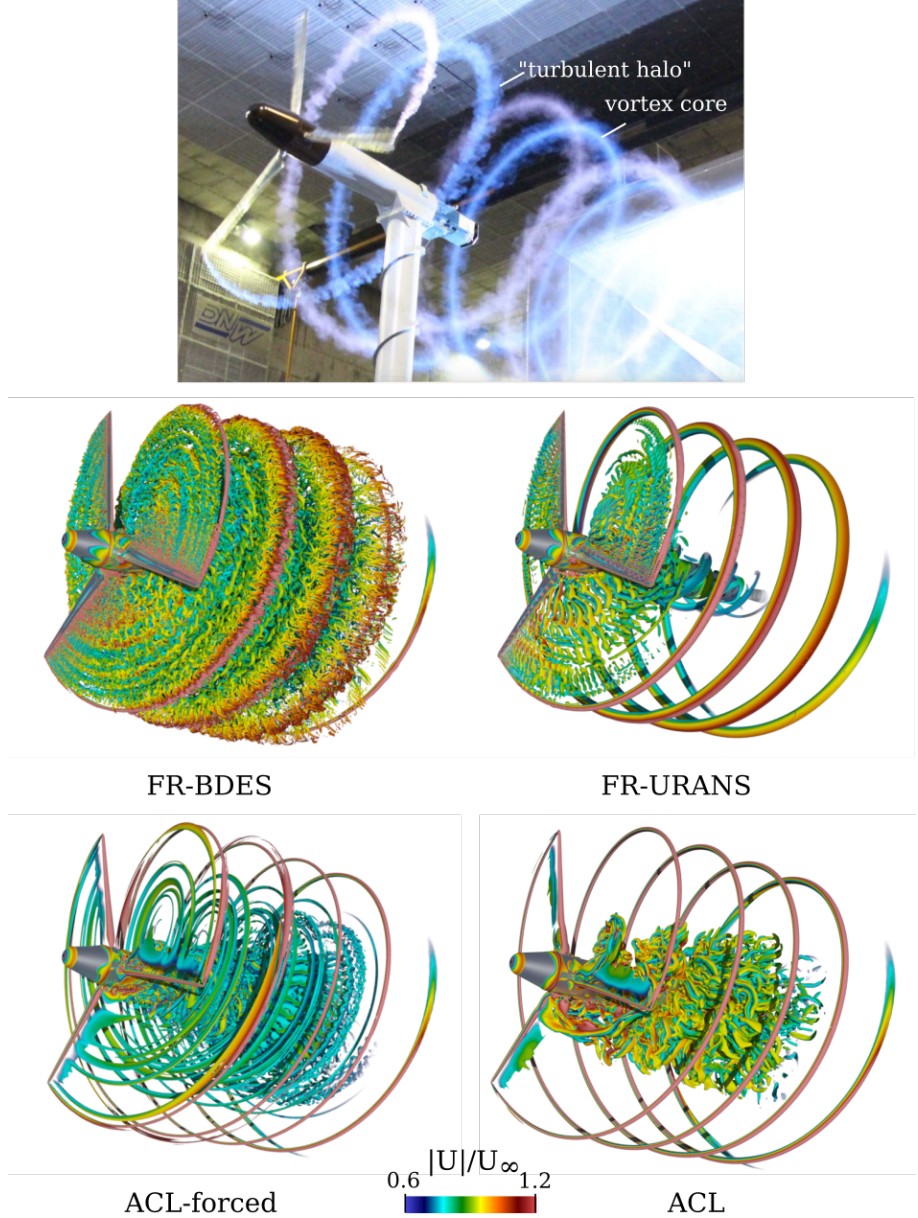

**Figure 4.** Flow Visualization of the wake of the MEXICO turbine. Comparison of experiments (Boorsma and Schepers, 2015) (top figure with courtesy from Schepers), with simulations using different turbulence and turbine modeling approaches.

In the URANS simulation the tip vortices are visibly larger than in the FR-BDES simulation and there are no secondary
vortices being created around the main vortex core. Similarly to the shape of the tip vortex, a coherent root vortex is formed, which is spiraling around the nacelle. The separated blade wakes are characterized by vortex shedding. This is the dominant mode, the one transient degree of freedom that can be resolved by the URANS method. The shedding frequency is seen





to increase with the rotor radius, expectedly, due to the increasing free-stream velocity and the decreasing thickness of the separated shear layer. This underlines the fact that only a single Strouhal number can be resolved and that any turbulence cascade, breaking down the eddies to smaller scales, is inhibited.

In the actuator line simulations, which were both carried out also with the BDES turbulence modeling approach[1], the tip vortices are significantly more concentrated than in the fully resolved URANS simulation. By nature, the actuator line is not able to resolve any flow-separation related shedding events in the blade wakes. Therefore, there is also no viscous shear layer. There is still a "pseudo-viscous" shear layer that originates from the momentum sources. This shear layer is though steady and less concentrated than what is observed in the blade-resolved simulations. Thus, no secondary vortices around the main tip vortex are formed, in contrast to the fully resolved BDES simulation. Indeed, this shear layer will depend on the user input model for the projection kernel. We use the formulation of Martínez-Tossas et al. (2017). Smaller kernels, e.g. directly proportional to the cell width may result in a sharper shear layer (in sufficiently fine grids) and potentially unlock these secondary vortices. However, this was not tested in the scope of the present work.

However, trailed vorticity caused by spanwise gradients in the bound circulation can be well captured, in general. These trailing vortices can be clearly seen in the mid-blade region and slightly inward of the main tip vortex of the ACL-forced simulation. These vortices do not appear in the standard ACL, at all, which already indicates that the underlying aerodynamic forces differ between the two simulations. Compared to the fully resolved URANS simulation there is also no coherent root vortex visible. It appears as if the wake in the inner region begins to mix, due to the shear inside of the wake. Overall, the standard ACL seems generate a larger and more complex root vortex pattern compared to its forced counterpart.

A closeup on the tip- and root will be looked at in the following sub-sections.

### 5.1.2 The topology of the tip vortex

The near field of the tip vortex is shown in Fig. 5 by visualizing the emerging vortices and the roll-up process again by means of a $\lambda_2$ iso-surface. In the fully resolved BDES simulation near the tip, the shear layers of suction side and the one of the trailing edge wrap around the very sharp vortex core. In the fully resolved URANS simulation the roll-up of the separated shear layers is present as well, however turbulent small-scale structures are completely absent. Although the diameter of the overall tip vortex seems qualitatively similar, no sharp vortex core can be identified. The pattern in the ACL simulations is similar for both, showing a vortex size comparable to the size of the overall tip vortex of the FR-BDES simulation, but involving no secondary vortices, as previously discussed. The only difference between both ACL simulations is the more pronounced vortex layer directly inboard of the main tip vortex resulting for the forced ACL. From that, the trailing vortex next to the tip vortex develops as seen in the overview visualization. As this vortex is also featured in the fully resolved simulations it is likely an effect of the local blade loading which will be discussed in detail in section 5.2.

---

[1]The shielding functionality will actually only treat the nacelle boundary layer in RANS mode. The wake region is totally treated in LES mode.





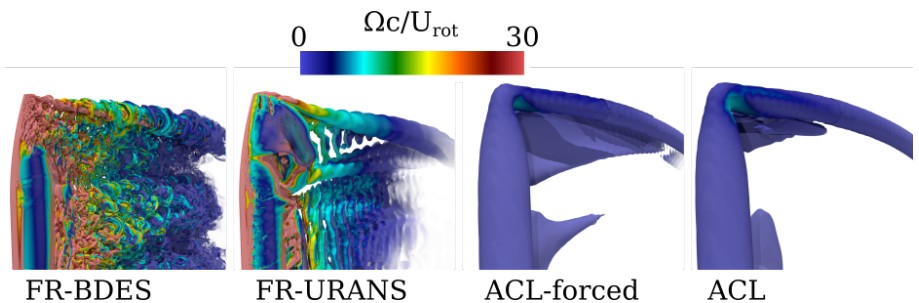

**Figure 5.** Comparison of the early tip vortex formation between for different turbulence and turbine simulation approaches.

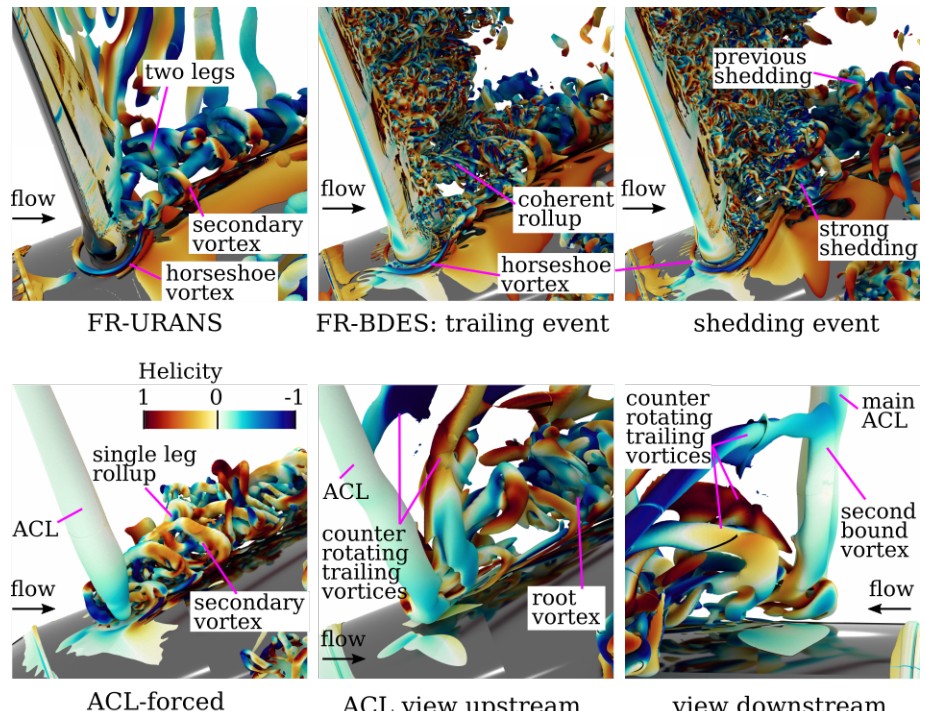

**Figure 6.** Comparison of the root vortex formation between fully resolved URANS and BDES simulations with the two actuator line simulations.





### 5.1.3 The topology of the root vortex

The general structure of the root vortex is compared for the simulations in Fig. 6. The contour color is the helicity to investigate
the sense of rotation of the vortices along the streamwise direction. In the FR-URANS simulation the root vortex consists of
two vortex legs which originate from the horse-shoe vortex that lays in the junction of the blade connector and the nacelle. The
helicity shows that at the horse-shoe vortex the two legs are counter-rotating. Farther downstream, they roll up into a single
vortex that rotates in the sense that there is a flow around the blade root from the pressure to the suction side. The vortex braid
itself spirals around the nacelle in the opposite sense of its self-rotation. In the course of the mutual interaction of the still
disjoined vortex legs, it can be seen from that secondary vortices are formed as a consequence of vortex stretching of the main
legs. Opposed to the URANS simulation the FR-BDES is capable of resolving the detailed flow phenomena going along with
the massive flow separation in the inner blade region. Two flow topologies can be observed which are depicted in the upper
right sub-figures, namely a trailing and a shedding topology. It is typical of vortex shedding phenomena that their intensity
modulates in time (Mockett, 2009). During weak shedding events a trailing topology similar to that of the URANS solution is
formed, although it is highly superimposed by smaller scale turbulent eddies. During strong shedding events, the lower leg is
destroyed and the detached flow is deflected in the outer direction into which also the roll-up of the root vortex is directed. The
roll-up itself is then also less coherent than in the case of the weak-shedding. For the ACL method where the forces are injected
from the FR-BDES, there is only one single vortex braid that spirals around the nacelle, as there is naturally no junction flow
that could form a horse-shoe vortex. Compared to the FR-URANS simulation, the ACL-forced simulation reveals smaller scale
turbulent structures. These vortices appear as the main root vortex is stretched by the boundary layer of the nacelle, and also
due to the velocity gradients caused by the rotor induction. The scale-resolving turbulence model facilitates the breakdown
into smaller and smaller eddies. In the FR-URANS simulation, this evolution is limited to the largest scales. The standard ACL
simulation involves a more complex vortex system in the root region, which shall be illustrated by means of a view upstream
and downstream of the actuator line. In the inner rotor region, a second bound vortex prevails that is enclosed by two trailing
legs. The outer one rotates in counter-clockwise direction with respect to the flow vector, whereas at the inner end, a counter
rotating vortex pair is formed. This vortex pair is deflected outwards. Inwards of it, the actual root vortex develops. Due to the
outward deflection of the whole vortex system, it is lifted from the nacelle surface at the beginning.

These observations can be explained in more detail when relating them to the blade loading, and particularly the distribution
of the circulation. Both will be discussed in the next section.

### 420   5.2 Blade loads, circulation and trailed vorticity

The time-averaged sectional blade loads are presented in Fig. 7. They are based on pressure integration in the normal and
tangential[2] direction of the local blade section and exclude the friction forces. In the experiments, the forces are integrated
from the pressure taps with the trapezoidal rule. Due to the sparse distribution of sensors in the aft-chord region, the trailing

---

[2]The tangential forces are defined locally relative to the airfoil chord. The tangential velocity in the wake used later on is defined tangentially in direction
of the to the rotor plane (positive in sense of the rotor sense).




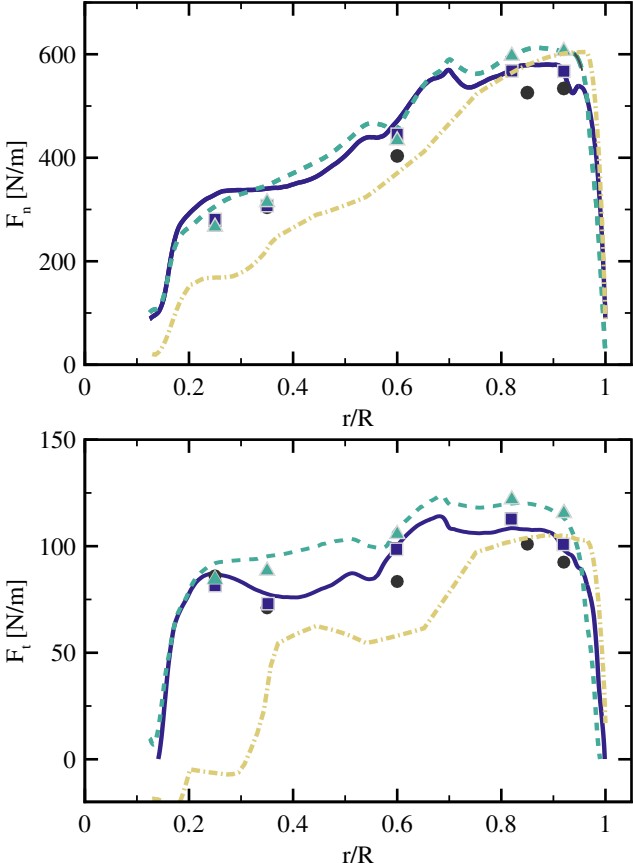

**Figure 7.** Sectional normal (upper graph) and tangential loads (lower graph). Experiment (●), FR-BDES with full pressure integration (——)
and with integration only at the sensor positions of the experiment (■). FR-URANS with full (– – –) and pressure taps integration only (▲),
ACL (–·–·).

edge point is approximated consistently to Parra (2016) by taking the average of the last available sensors in the chord direction

from the pressure and suction side. In the simulations, two different evaluation methods are compared: The first calculates the
forces by integration of pressure over the entire surface. The second utilizes only the data points at the location of the pressure
taps in the experiment. In the actuator line method, the forces stem directly from the respective actuation nodes, where the
BEM is evaluated.

    In general, it becomes apparent that the simulations tend to overestimate the normal and tangential loads, particularly in

the outer blade region. However, it must also be noted that the deviations in the inner and mid part of the rotor are at least
partly a matter of the post-processing. It can be shown that by evaluating the loads solely based on the positions of the pressure
taps, an overall improvement of the agreement is achieved, yielding an almost perfect match for the two inner sections. At
$r/R = 0.6$, the overestimation of the normal loads might be due to the fact that the actual geometry and thus blocking effect of
the zig-zag tape is not considered in the simulations. It was suspected within the MexNext III project that the latter may cause



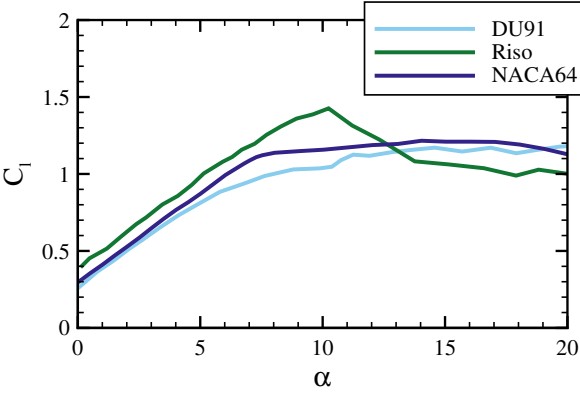

**Figure 8.** Measured 2D lift polars of the DU91-W2-250, the Risø-1-21 and NACA64-418 airfoils (reproduced from Schepers et al. (2012)).

an overtripping in the experiment particularly in the Risø section and consequently lead a larger extent of the flow separation. In the region of the blending between the Risø and the NACA 64 airfoil the URANS and the BDES simulation show a very similar behavior in the mean loads, namely strong spanwise gradients. These are linked with gradients in bound circulation, which will be detailed in Fig. 10. They are in equilibrium with the two trailed vortices being created in the transition zones between the airfoils. Such strong gradients in the loads were not visible in any previous simulations on this operating point, but might be eventually captured by the substantially finer grids employed within the present study. As there are no experimental data in this region, a comprehensive validation is not possible. At the two outer sections, the overestimation of the normal loads is associated with the problem of too small a tendency of flow separation which is a typical tendency of the employed $k$-$\omega$ SST turbulence (background) model (Matyushenko and Garbaruk, 2016). The FR-BDES nevertheless shows an improvement compared to the URANS approach. The advantage of the BDES is especially pronounced in the tangential forces, which are significantly influenced by the pressure drag of the separated flow at each profile cut. This suggests that generally the extent of the flow separation seems reasonably predicted. Only again in the outer sections there is a slight overprediction of the tangential forces corresponding to the separation area predicted there being too small.

As stated in the outline, it is a main objective of the present study to validate the actuator line method for one case in absence of the uncertainties in the underlying BEM or the polars. Therefore, the mean blade loads from the FR-BDES simulation are used for the ACL simulations in a forced mode and injected as momentum sources into the Navier-Stokes equations. In the standard formulation, where the model calculates the forces via the BEM, normal and tangential loads are significantly lower than in the fully resolved counterparts, particularly in the inner rotor region. It should be noted that 2D polars were applied without additional corrections or a model for rotational augmentation. Hence, the impact of that effect seems to be significant, which is supported by the fact that very similar force distributions were obtained by all partners working with lifting line methods and did not employ a rotational augmentation model, while other BEM codes with respective models showed higher loads and better agreement to the measurements (Boorsma et al., 2018). In the transition zones between the airfoils, the force



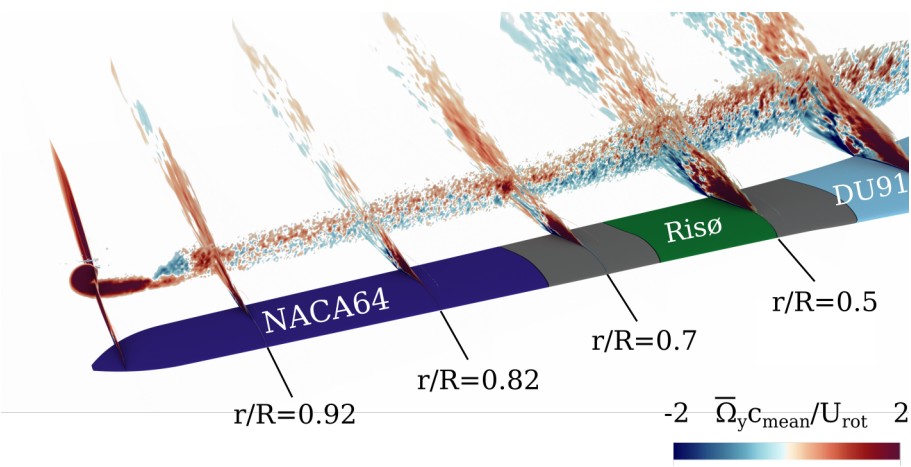

**Figure 9.** Time averaged chordwise vorticity in the separated wake indicating trailing vortices in the transition regions between the airfoils. Simulation FR-BDES.

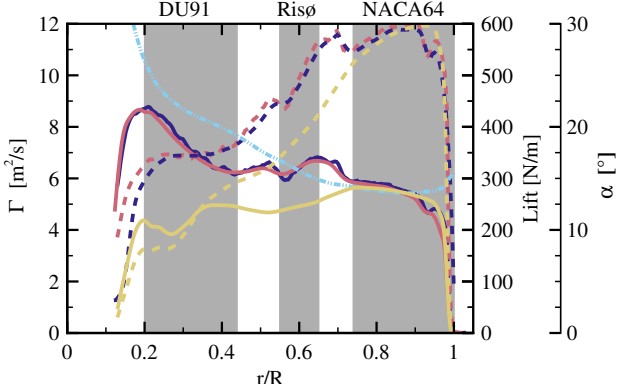

**Figure 10.** Radial distributions of the bound circulation $\Gamma$ for the FR-BDES (——), ACL-forced (——) and ACL (——). The sectional lift forces are included as calculated with the Kutta-Joukowski theorem for ACL-forced (- - -) and ACL (- - -) and compared to the surface loads from FR-BDES (- - -) using the angle of attack distribution (- · -) extratced from FR-BDES. The grey patches indicate the locations of the different airfoil types.





gradients are much smaller. First, there are no measured polars in the transition zones. Therefore, the BEM interpolates them from the adjacent airfoils, i.e. between the DU91 and the Risø for the inner transition zone and between the Risø and the NACA64 for the outer zone. Comparing the measured polars (see Fig. 8), it can be seen that the stall onset of the airfoils
is different. The Risø airfoil has a higher $C_{L,max}$ and a harder stall compared to the DU91 airfoil and also compared to the NACA64 airfoil. However, looking at the angles of attack at which the turbine operates (see Fig. 10), it can be seen that in the $0.45 r/R <= 0.75$ region the angles are between $20°$ and $13°$. This means that for the ACL with 2D BEM calculation, all polars are effectively in the post-stall regime where very similar lift values are obtained. Thus, the spanwise gradients between the individual airfoils in the transition zones become small. On the other hand, in the blade-resolved simulations, three-dimensional
effects such as cross-flow cause the airfoils not to operate in the flat post-stall regime and the different behavior of the airfoils in the stall region comes into play more than in the ACL with BEM forces. In the outer blade region, $r/R > 0.75$ a comparable agreement to the fully resolved simulations is achieved.

An evaluation of the bound circulation is shown in Fig. 10. It aims at helping to better understand the formation of trailing vortices and more importantly, to supply evidence on objective 3 stated in the outline, namely to clarify, whether the same
wake of a fully resolved simulation is obtained when imposing their forces into the ACL. For this purpose, the circulation distributions evaluated in the flow field of both simulations shall be compared. If they do agree well, it can be expected, that also very similar mean velocities are obtained in the wake. Moreover, the validity of the Kutta-Joukowski theorem shall be verified by comparing lift, recalculated using the evaluated bound circulation in the ACL-forced with lift from the surface loads of the FR-BDES simulation.

In the graph, circulation is displayed as solid lines, lift as dashed lines. The FR-BDES, the forced ACL and the standard ACL method are given in blue, rose and sand colors, respectively. The circulation was determined from the time averaged velocity fields extracted on a cylindrical surface with $0.37\,\mathrm{m}$ radius around the blade by integrating along closed circular curves in each section of the hull using the relation $\Gamma = \oint \boldsymbol{u} \cdot d\boldsymbol{s}$.

Firstly, it can be observed that the fully resolved simulation agrees very well with the forced ACL. This also holds for
the comparison of lift, where the recalculated one from Kutta-Joukowski theorem almost perfectly matches the distribution evaluated from the surface of the FR-BDES simulation[3] (besides of some minor deviations in the very inboard region). This gives clear indication that imposing forces stemming from fully resolved simulations directly into the actuator line results in the same circulation, even in separated and consequently, it can be expected that also a very similar induction and mean velocity distribution is obtained in the near field of the blade.

When comparing the standard ACL model to that, a significantly smaller circulation is generated, particularly in the inner blade region. This already implies a much lower induction of the rotor in that area and emphasizes the necessity of taking into account corrections for rotational effects in BEM or lifting line codes, when flow separation is encountered in the root region of the blade.

---

[3]The required angle of attack was evaluated using the method of Jost et al. (2018), by averaging the axial and tangential velocities on the hull surface in the same way as calculating the circulation.



As the gradients in the circulation behave proportional to the trailed vorticity, they are looked at more closely to explain the observed trailing vortices in the flow visualizations. In the FR-BDES it is difficult to identify these vortices in the instantaneous iso-surfaces of $\lambda_2$, as they are heavily overloaded by small-scale turbulence. Therefore it shall be referred to Fig. 9, where the temporal mean of the vorticity component in the chordwise direction is depicted.

The overall largest gradients appear at the root and the tip, in all simulations, and are in equilibrium with the root and tip vortices, respectively. However, as already suspected from the loads, significant local gradients appear in the transition zones to and from the Risø airfoil, but only in the FR-BDES and the ACL-forced simulation. Those correspond to the trailing vortices in the mid part of the blade, which are not present in the standard ACL, as observed in Fig. 4. Moreover, close to the tip at $r/R \approx 0.95$ there is a small dip, slightly more pronounced in the FR-BDES than in the ACL-forced simulation that is represented by an additional trailing vortex inboard of the main tip vortex. This dip is not present in the standard ACL showing and thus, also no additional vortex evolves at the respective position. On the other hand, this simulation features a spot of negative and then positive gradient in the inner region $r/R \approx 0.2 - 0.35$ not appearing in the other simulations. This explains the additional trailing vortices observed in Fig. 6. The root vortex and the outer trailing leg that both exposed a negative helicity are associated with the positive circulation gradient, whereas the vortex slightly outboard of the actual root vortex that revealed a positive helicity corresponds to the negative circulation gradient.

## 5.3 The wake of the blade

The wake of the blade mainly reflects the induction effects caused by the aerodynamic forces discussed before. However, also viscous effects stemming from the boundary layers, or on a larger scale, separated shear layers are visible in the blade wake. With the actuator line simulations imprints of the bound vortex result also in shear layers that create a similar pattern in the flow field as the viscous and separated shear layers, although they origin from the integral aerodynamic forces of the blade section.

### 5.3.1 Comparison of flow structures with PIV

In the following, a closer look is taken at the instantaneous flow fields extracted in planes with azimuthal increments $\Delta\theta = 13, 20, 40\,°$ downstream of the blade. These are shown for the axial, tangential and radial velocity components in Figs. 11 – 13, respectively, comparing all simulations with the PIV data. The locations of the PIV sheets are drawn also into the graphs of the simulations, where the regions outside of the sheets are made translucent. In general, there is a spatial shift between the PIV sheets and the slices of the simulations extracted at the nominal azimuthal positions. Schulz et al. (2016) found the cause of this offset in a misalignment of the coordinate systems by around $\propto 0.045\,m$. In the contour plots the position of the radial velocity traverses, which will be discussed in the next section are displayed. They are nominally to be extracted at $x = 0.3\,m$. In the experiments the position is corrected to $x = 0.345\,m$.

At the position $\Delta\theta = 13\,°$ downstream of the blade, where the separated flow field has just entered the evaluation plane, the experiments show alternating cells of high and low axial velocity along the mid and outer blade region. The bubble-shaped areas of high tangential velocity emphasize the alternating sizes of the separation area and indicate the presence of stall cells.





In the inner region, where the blade is just entering the plane, the axial velocity is negative, as consequence of the downwash of the blade and the generation of lift. This goes in hand with a negative tangential velocity, due to the large circulation and results in a deflection of the streamlines into the plane. Directly at the root the spot of positive velocity indicates the formation

of the root vortex that pushes the flow on the suction side radially outward (cf. radial velocity in Fig. 13). At the blade tip, the corresponding tip vortex has already formed, since the evaluation plane is situated some chord lengths downstream of the blade. The circumferential velocity around the vortex axis is reflected in the axial and radial velocity (related to the rotor). The axial velocity of the tip vortex is aligned approximately with the tangential direction of the rotor. However, the tilting of the vortex helix with respect to the evaluation plane, which is not negligible due to the small TSR of the rotor, leads to a spot of

positive tangential velocity at the upper border of the vortex and a negative spot on the bottom side.

Comparing now the simulations with the PIV measurements in the first plane, it can be found that the FR-BDES simulation yields a structurally very similar flow pattern, namely alternating patches of axial velocity in the mid and outer part of the blade. Those show up as bulbs carrying positive tangential velocity and emphasize the modulation of the separated flow of the blade. Compared to the experiment, the predicted magnitude of the velocity is slightly higher in the mid blade region, while

quantitatively similar levels are obtained in the outer sections. Regarding the sizes of the turbulent eddies, those are even finer in the FR-BDES than in the PIV. This is because of a higher spatial resolution in the near-field of the blade in the simulations, which is $1.4\,\mathrm{mm}$ for the inner half of the rotor and $0.7\,\mathrm{mm}$ in the outer portion, compared to a maximum PIV resolution given as $2.14\,\mathrm{mm}$ (Boorsma and Schepers, 2015). Compared to that, the velocity field in the FR-URANS simulation is less fine grained, although the main alternating vortex structures in the mid and outer blade region are also present as can be retrieved

especially from the axial velocity field. In the ACL simulations, by definition no viscous blade aerodynamics can be taken into account. Therefore, the detailed and transient separation features and their imprint into the very near wake of the blade cannot be resolved. Indeed, for a fairer comparison of ACL, the time-averaged fields from PIV and the full-resolution simulations would need to be used. The very basic flow patterns that mainly depend on the integral airfoil forces are though captured as can be seen in the qualitatively similar shape of the tangential velocity. Here, the standard ACL shows a more homogeneous

flow field in the mid and outer blade region compared to the forced ACL. This is aligned with the fact that the forces and their gradients are much smoother in the standard ACL compared to the forced variant, which is in accordance to the previously made observations made on the generation of the trailing vortices.

In the inner blade region the flow field is very well predicted by the FR-BDES simulation. This applies to the deceleration of the flow in the axial direction, its strong deflection in the negative tangential direction, and its redirection in the positive radial

direction. The first two effects are due to the strong circulation of the blade in the inner portion, whereas the region of positive radial velocity represents the suction side part of the root vortex that is building up. The spot of positive tangential velocity near the vertical line that marks the extraction of the radial traverse shows the tail of a preceding shedding event as described in the flow visualisation of Fig. 6. Apart from the absence of small-scale turbulence, the FR-URANS simulation provides a very similar picture compared to the BDES simulation. Merely, the range of influence of the root vortex in all velocity components

is narrower than in the FR-BDES and the experiments. Accordingly, there is no vortex shedding in the inner blade area. The ACL-forced simulation shows a similar pattern in the root vortex region for the tangential and radial components as the FR-





BDES simulation, from which it inherited the forces. Overall, however, the flow deflection in the tangential direction appears to be less pronounced than in the fully resolved simulations and the experiment. Relative to this, however, the deviations of the ACL, in which the forces are calculated internally by the BEM, are significantly larger. The deceleration in the axial direction is smaller and especially the deflection of the flow in the negative tangential direction is completely absent. Instead, the secondary vortex system shown in Fig. 6 is forming. This flow pattern correlates in detail with the clearly underestimated blade loads in normal and tangential direction compared to the experiment, as well as the associated distinctly reduced circulation in the inner blade area discussed in Fig. 10.

In the tip region, the roll-up process of the vortex occurs similarly in all simulations. The best agreement with the experiment is obtained with the FR-BDES simulation. In the FR-URANS simulation the tip-vortex is already deformed and more oval than in the experiment. The ACL simulations predict the shape correctly, although the initial strength seen in the peak velocities slightly differs among forced and standard ACL. There are differences in the tangential velocity pattern of the tip vortex. In the FR-BDES the shape is very similar to the experiment, however, with slightly stronger peak values in both directions. Compared to that the positive spot in the upper border of the tip seems to be highly pronounced in the FR-URANS, but also in the forced ACL. On the other hand the negative velocity spot at the lower end of the tip vortex is overly pronounced in the standard ACL. Especially when tracking the evolution in the planes further downstream, the strength of the tangential velocity in the tip vortex seems to depend on the strength of the shear layer being rolled up. In the forced ACL this layer of positive tangential velocity remains coherent. The roll-up of the tip vortex stretches the shear layer and thereby transports high values into a concentrated spiral around the main tip vortex (significantly stronger than in the experiment). In the ACL with internal force calculation this layer is more blurred. The reason might lie in the bi-directional coupling of the forces and the wake flow. Hence, variations in the flow field will vice-versa cause variations in the forces and lead to an overall thicker and less concentrated shear layer. Of course, this mechanism does not exist in the variant with constantly applied forces. By comparison, the shear layer in the fully resolved simulations includes the viscous effects of the separated blade wake. Thus, especially with the scale resolving turbulence modeling, the shear layer mixes and becomes less coherent than in the ACL-forced simulation. Consequently, also a less pronounced and less concentrated spiral of positive tangential velocity is rolled-up around the tip vortex. This is also in overall good agreement with the PIV measurements. In the FR-URANS simulation, though, a higher blurring is visible due to the dissipating nature of the underlying turbulence modeling.

When looking at the propagation of the separated wake in the mid and outer part of the blade (note that the PIV results are not temporally correlated to the planes before), the turbulence pattern is again very similarly predicted in size and shape by the FR-BDES. The mushroom-like structures have become larger during the development from the previous slice. At the position $\Delta = 40°$ relative to the blade a large-scale turbulence bale has developed in the wake of mid-blade region, a narrow turbulence band in the inboard area and a relatively homogeneous wake in the outer part of the rotor. Regarding the resolution of small-scale structures good visual agreement is found only in the inner half of the rotor. Farther outboard, however, where the distance of the slice to the blade has increased, the wake is already in the coarser grid level and therefore, the eddies become blurred. In the FR-URANS simulation the changes along the different evaluation steps are primarily related to inviscid induction effects that govern the motion and interaction of the largest modes being resolved. Any other turbulence-related cascade of



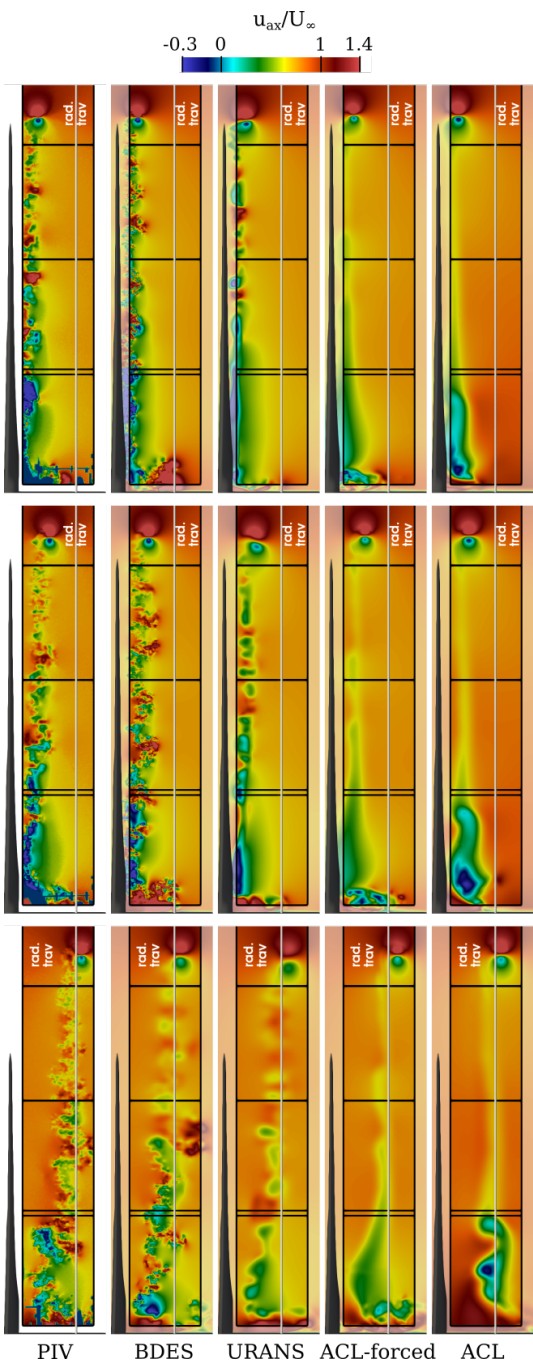

**Figure 11.** Evolution of axial velocity in the blade wake in planes $13°$, $20°$, $40°$ (from top to bottom) downstream to the blade. Experimental results from (Boorsma and Schepers, 2015).

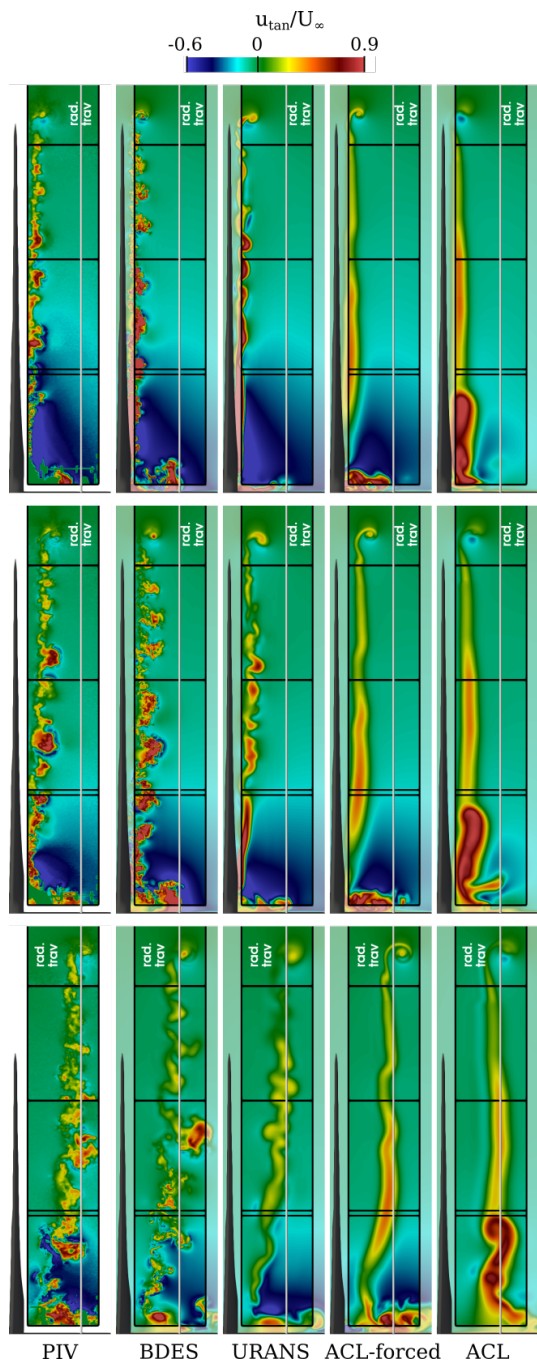

**Figure 12.** Evolution of tangential velocity in the blade wake in planes $13°$, $20°$, $40°$ (from top to bottom) downstream to the blade. Experimental results from (Boorsma and Schepers, 2015).



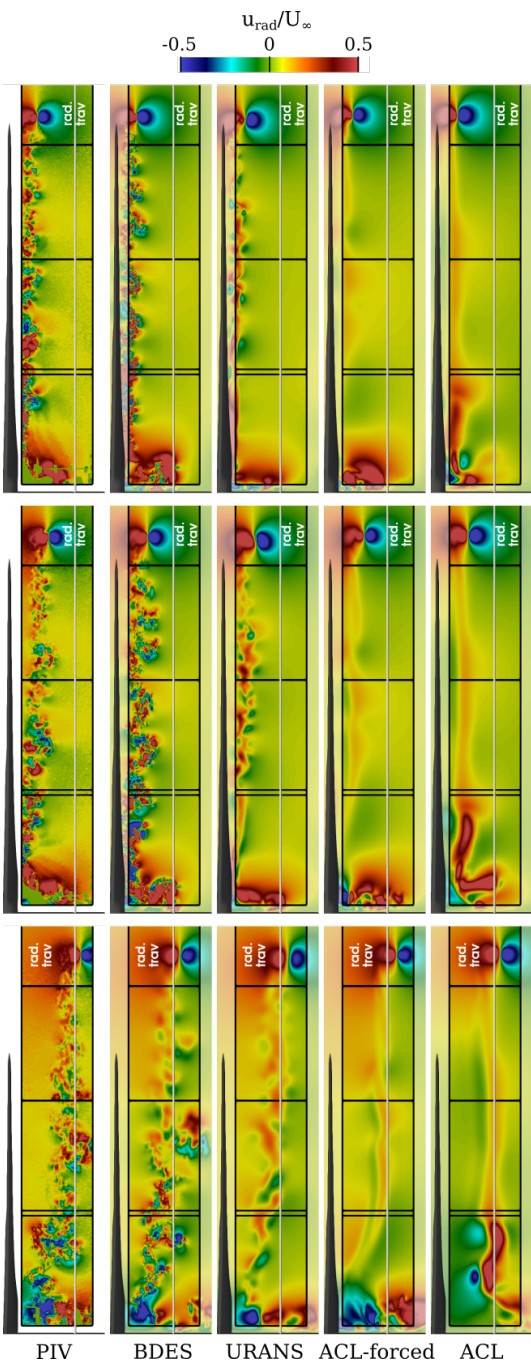

**Figure 13.** Evolution of the radial velocity in the blade wake in planes $13°$, $20°$, $40°$ (from top to bottom) downstream to the blade. Experimental results from (Boorsma and Schepers, 2015).





vortex breakdown is diffused by the turbulence model. This the typical and expected behavior of the URANS approach and commonly referred to as the "spectral gap". Likewise, the ACL flow field is also primarily driven by inviscid effects (except for the wall-bounded part near the root). However, in comparison to the URANS simulation the diffusion is minimized by using a
scale-resolving turbulence model. Since there are no external perturbations, like gusts or inflow turbulence, the wake remains rather stable in the near field of the blade. First deformations are though initiated with the ACL-forced simulation at $\Delta = 40°$ due to the high gradients in the bound circulation of the ACL-forced simulation initiate a deformation wake.

Turning finally to the propagation of the flow in the root region, the details of the described flow topology in Fig. 6) can be found also in PIV snapshot at $\Delta = 40°$. The plane is located temporally in between of two strong shedding events and thus,
capturing a trailing event in which the root vortex is rolled up. This can be retrieved from the areas of alternating sign in the axial and radial velocity components. Unlike the tip vortex, which is almost circular, the root vortex becomes elliptical by the presence of the shear in the boundary layer of the nacelle. An important detail to be noted is the small patch of positive tangential velocity at the edge of the PIV plane observed in both the FR-BDES simulation and in the experiment. A close examination of the $\lambda_2$ isosurface in Fig. 6 has shown that the evaluation plane is intersecting the offshoots of the previous shedding event.
The fact that this spot is very similarly found in the experiments gives a clear indication on the actual occurrence of the described alternations of the root-vortex topology. In the FR-URANS and ACL forced simulations, the root vortex develops quite similarly to each other. However, as already noted for the first evaluation level, it is somewhat flatter in the FR-URANS simulation. Lastly, the most significant deviations pretend in the standard ACL simulation. In particular from the radial velocity component it can be clearly seen how the two additional inboard vortices previously described in Fig. 6 are developing. Those
also inhibit the classical velocity pattern expected of the root vortex.

### 5.3.2  Radial traverses

The radial traverses of the velocities are extracted $0.13R$ downstream of the rotor blade at different (averaged) phase angles relative to the blade (see also vertical lines depicted in the Figs. 11 – 13). As stated before the evaluation position in the experiments is corrected according to Schulz et al. (2016) to $0.153R$. Moreover, the mean values were calculated from 31
temporally uncorrelated images. In the simulations, phase-averaged results are presented. A comparison of simulations and measurements for all velocity components at the relative azimuth positions $\Delta\theta = 20, 40, 60°$, as well as a weighted average over the whole $120°$ period is shown in Fig. 14. The latter was used as the main benchmark for the code-comparison rounds in the MexNext III project (Boorsma et al., 2018). It was calculated by multiplying each traverse by the relative azimuthal increment between two positions. In the measurements, the error-bars shown as grey background denote the standard deviation
of the velocity. Note that for the azimuthal average, the weighting is applied equally to the standard deviation.

The azimuthal average of the axial velocity is shown in the upper left graph of Fig. 14. Overall, good agreement of simulations and measurements can be observed. The only significant deviations appear with the ACL simulation in the inner region, where induction is generally too small. In the outer part all simulations agree well with the measurements. It is worth noting that the previously observed overestimation of the loads in the outer rotor area is hardly discernible in the velocity deficit. The
drop in the inboard region between $r/R = 0.2 – 0.4$ is seen to be caused by the large normal-force coefficients in this area.



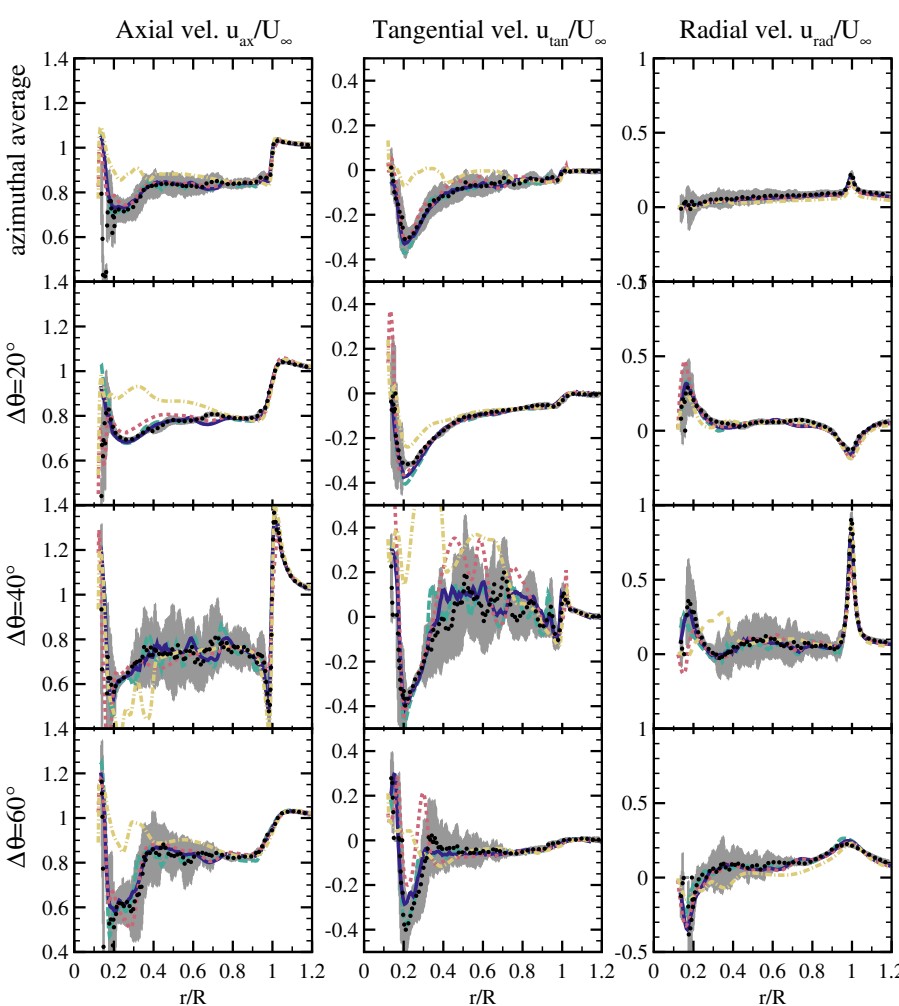

**Figure 14.** Radial traverses of the axial, tangential and radial velocity, evaluated $r/R = 0.13$ downstream of the rotor plane at different azimuthal positions relative to the blade ($\Delta\theta$) as well as azimuthally averaged. Experiment (Boorsma and Schepers, 2015) (●) with standard deviation as grey background. Simulations: FR-BDES (——), FR-URANS (– – –), ACL-forced (······), ACL (–·–·–).





Note that for the design case at tip-speed ratio $\Lambda = 6.7$ this dip did not occur and also the blade-normal loads did not show an increase in that section. Hence, this effect is associated with the rotational augmentation of lift in the inboard region of the blade. In the ACL simulation, where the forces are internally calculated with the BEM, this effect is not included. Likewise, the underprediction of the normal loads and the associated reduced circulation in that region leads to a significant underprediction

of the axial induction in that region. In the ACL with imposed forces the agreement is very good on behalf of the azimuthally averaged wake. Only in the early stages of building up the wake (cf. $\Delta\theta = 20°$) the flow is less retarded than in the experiment. At the plane $\Delta\theta = 40°$ the separated flow and also the tip vortex reaches the evaluation traverse. Excellent agreement is achieved for all simulations. However, it must be noted the correction of the evaluation position in the experiments is crucial to correctly capture the large gradients around the tip vortices. The separated flow which is stemming from the mid and outer

blade region, yields expectedly strong fluctuations. In the inner portion the fluctuations are lower, since this part of the separation area travels slower. All simulations closely match the measured velocities of the PIV. Only in the inner part the secondary vortex system evolving in the standard ACL leads to a downward excursion in the velocity. At $\Delta\theta = 60°$ the separated area of the root region has arrived imposing large fluctuations, whereas the fluctuations in the outer area have diminished. Again, very good agreement is obtained for all simulations, except for the case of the standard ACL in the inner part of the blade.

Turning to the tangential component $u_{tan}$, its profile is similar for all evaluated azimuth positions, showing a distinct minimum at around $r/R = 0.2$ followed by a rise towards the tip. Generally, the values are supposed to be negative as the wake is rotating in the opposite direction of the rotor. The only exception, where positive values are obtained is when the separated wake flow of the blade reaches the evaluation line (see $\Delta\theta = 40°$). As due to the principle of "actio and reactio" the rotor can only be driven if it sets the wake into counter-rotation, it is intuitively clear that the positive tangential component caused by

the flow separation reduces the turbine power. The strong dip in the inner blade region, which was visible as the dark-blue contour in Fig. 12 is again the consequence of the high bound circulation in the root region boosted by the effect of rotational augmentation. Indeed, pressure coefficients of up to $C_p \approx -8$ (see code comparison in Boorsma et al. (2018)) were obtained on the suction side of the blade. In the standard ACL without respective correction for the rotation effects, the associated deflection of the flow is significantly less pronounced. All other simulations yield very accurate simulations in that region. At $\Delta\theta = 40°$

the ACL-forced simulation tends to predict too positive a tangential velocity in the mid blade region. This overprediction is even more pronounced in the standard ACL simulation. It implies that the shear layer in the ACL simulations stemming from the injection of body forces is somewhat too concentrated compared to the ones of the fully resolved simulations that account for the effect of turbulent mixing. The force injection could be further tuned by employing different strategies for mapping them into the flow field (Cormier et al., 2021; Churchfield et al., 2017). In the tip region all simulations capture the change

in sign of the tangential velocity from negative to positive values in position and magnitude. This implies that the roll-up and tilting of the tip vortex relative to the evaluation plane is predicted correctly. At $\Delta\theta = 60°$, where the traverse is cutting the root vortex, the FR-URANS simulation yields the best agreement, whereas its BDES counterpart slightly underestimates the dip. In the forced ACL the change in sign is too concentrated and not predicted as in the measurements. As already seen from the distribution of the axial velocity, the flow deflection in tangential direction in the rotor blade inner area is only insufficiently

represented in the ACL with internal force calculation.





In contrast to the other components, for which the general distribution was the same across all azimuth positions, this is not the case with the radial component. At $\Delta\theta = 20°$, which is before the arrival of the separated flow, the root vortex induces an outward velocity and correspondingly, the tip vortex a negative component. In between, an overall outward motion prevails. Opposed to the axial velocity, the ACL-forced simulation is in good agreement with the experiments. At $\Delta\theta = 40°$, the tip vortex has just passed the traverse and the root vortex not yet arrived, both vortices induce a positive radial velocity as also shown in the lower rows of Figs. 11- 13. The simulations very accurately capture the distinct peak in the radial velocity. The excellent agreement also holds for the azimuth of $60°$, except for a slight underprediction visible in the standard ACL. At this time, the root vortex has passed and induces an inward velocity component. In the tip region, the traverse departs the influence region of the tip vortex. Returning to the azimuth-averaged graph (note the changed axis scale), it becomes clear that most of the local effects are averaged-out. What remains is an overall outward flow, which is slightly lower with the standard ACL compared to the other simulations. This is associated with an overall smaller thrust coefficient of the rotor that also leads to a reduced deflection of the outer flow around the rotor wake.

### 5.3.3 Summary on the development of the blade wake

The following conclusions can be drawn from the velocity field in very near wake of the blade:

1. The general trend in the velocity field is well predicted by all fully resolved and actuator line simulations.

2. The scale-resolved simulation on a highly resolved grid provides a very detailed view into the physics and the development of turbulent structures.

3. The URANS approach is though sufficiently accurate in predicting the mean induction effects, even in separated flow.

4. The standard actuator line with internal force calculation and no model for rotational augmentation underpredicts axial and tangential induction in the inner part of the blade.

5. The induced secondary vortex system at the inner blade region (not the actual root vortex) in the standard ACL has no physical basis and is a matter of the falsely predicted forces and gradients.

### 5.4 The wake of the rotor

In the MexNext projects, the near wake of the rotor was evaluated by means of axial velocity traverses. The corresponding evaluation plane is located $90°$ downstream of the rotor blade. In the following again first a qualitative comparison between the different simulations shall be made based on the velocity contours in that plane, before the traverses are cross-plotted.

### 5.4.1 Flow structures in the rotor wake

The velocity distributions of the PIV sheets are presented together with the predictions of the simulation in Fig. 15. Unfortunately, the axial placement of the sheets for vortex tracking was adapted to the pitch of the wake helix at the TSR $\lambda = 6.7$





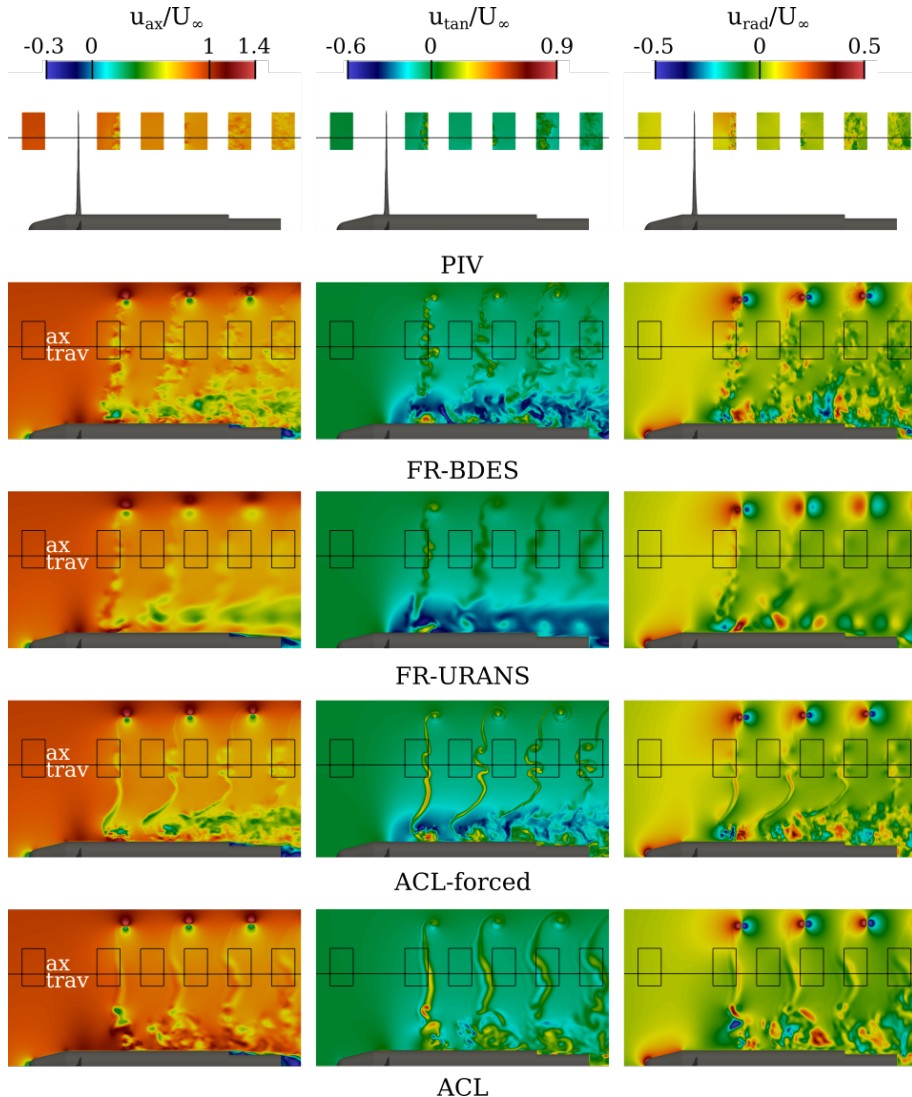

**Figure 15.** Comparison of the flow field in the wake of the rotor in a plane located $90°$ downstream to the blade. PIV measurements from (Boorsma and Schepers, 2015).

case, for which the distance between two consecutive tip vortices is smaller. Thus, in the present case, the wake of the blade cannot be entirely captured by the PIV sheets. In the FR-BDES simulation the mid and outer area of the wake shows that the coherent structures disintegrate with increasing downstream distance. In particular the velocity in the tangential direction shows that the wavy pattern that was characteristic for the wake of the blade breaks down towards homogeneous turbulence. This development is also confirmed by the experiments. However, due to the coarser computational grid in the wake compared

to the PIV resolution, the simulated turbulence fields are more diffused. In the FR-URANS simulation, it can be clearly seen that any scale-resolution is inhibited. The initially wavy shape in the tangential velocity is maintained. It appears to be only de-





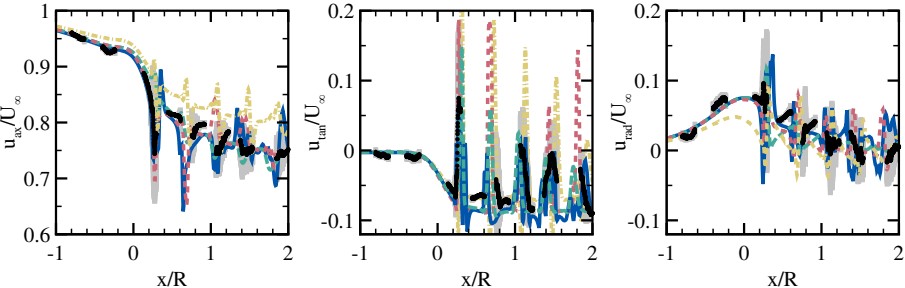

**Figure 16.** Axial traverse of the axial, tangential and radial velocity components, evaluated at $r/R = 0.67$. Experiment (Boorsma and Schepers, 2015) (●), Simulations: FR-BDES (——), FR-URANS (- - -), ACL-forced (·····), ACL (-·-·-).

formed by mutual induction effects. With increasing downstream distance these structures are increasingly diffused. Compared to the results of the fully resolved simulations, the ACL-forced simulation predicts significantly sharper velocity gradients in that region of the wake compared to the URANS flow field. This is firstly due to the turbulence modeling by using the same

approach as the FR-BDES and secondly because of the stronger gradients in the shear layer generated by the body forces. With increasing downstream distance it can be observed that the trailing vortices in the mid region of the rotor start becoming unstable. Compared to that the standard ACL reveals a less wavy pattern which is also more diffusive than in the ACL-forces. As already discussed before, the missing waviness is due to the absence of any trailing vortices in the mid region of the rotor and the increased diffusivity may be rooted in the bi-directional coupling of forces and wake that could have a damping effect

on the imposed body forces.

Focusing on the inner region, the root vortex development can be traced by the alternating signs in the radial velocity component $u_{rad}$. In all simulations, but the standard ACL, they develop similarly. It can deduced that its convection velocity is lower compared to the tip vortex due to overall smaller axial velocity in the inner region of the wake. Due to the shear which is enhanced by the boundary layer around the nacelle the root vortex starts mixing with the flow in the center. In the

FR-BDES simulation it also mixes with the resolved turbulence of the separated blade wakes. Along the axial direction the thickness of the mixing layer increases. When reaching the position of the step in the nacelle geometry, no coherent vortex is visible anymore. In the FR-URANS simulation the root vortex remains coherent without being mixed with the surrounding, but it is blurred out during the propagation. Opposed to that, the ACL-forced simulation predicts a very similar topology as its FR-BDES counterpart, showing also a disintegration towards small scale turbulence. As there are no resolved eddies in the

blade wakes, the mixing is slightly slower. In the standard ACL the wrongly predicted development of the root vortex with its induced secondary trailing vortices together with the smaller induction in the inner part of the rotor reduce also the radial shear in the wake. Therefore, the mixing with the core flow is also less pronounced.

The latter observations on the actuator line method, namely that the wake remains more stable in undisturbed conditions, but reaches a very similar topology as fully resolved simulations when it is disturbed like here due to the shear was already stated

by Troldborg et al. (2015), who compared the ACL with fully resolved simulations in uniform and turbulent inflow.





### 5.4.2 Axial traverses

A quantitative comparison of the velocity fields is conducted based on the axial traverse crossing the mid area of the wake at $r/R = 0.67$ (see Fig. 16). The PIV measurements are phase-averaged and the grey background shading indicates the standard deviation. In the simulations, the instantaneous distributions are plotted. The axial velocity implies the typical gradual

deceleration of the flow when approaching the rotor, due to its blockage effect, and a further, more distinct retardation after passing the rotor plane being caused by its axial induction. The induction factor is only $a_{ax} = 0.13$ at $r/R = 0.8$, which results that the average of the velocity is still around $75\%$ of the inflow. In the simulations, the general shape of the blockage and the retardation of the flow in the wake are well predicted. Only the standard ACL underpredicts the blockage and the wake deficit, due to the corresponding underprediction of the loads in the inner and mid portion of the blade.

Other than that, it can be observed that the axial velocity drops periodically when crossing the detached wakes of the blade. In the tangential velocity this effect is even more pronounced as positive excursions. In both ACL simulations these peaks are overpredicted, as the shear layers induced by the body forces are predicted too strong and too stable. This decay is more realistically captured by the fully resolved simulations. In the BDES simulation the decay is a consequence of the resolved turbulent mixing, whereas in the URANS it is caused by the modeled turbulent viscosity.

The general sense of the tangential velocity primarily indicates the rotation of the wake being opposed to the rotation of the rotor. It is interesting to note that the wake rotation already initiates slightly upstream of the rotor. In the wake it then remains effectively constant. All simulations slightly overpredict the magnitude of the wake rotation. This could be attributed to the overprediction of the loads in the tangential direction at that radial position.

Turning finally to the radial velocity shown in the right panel of Fig. 16, it is characterized by a positive velocity upstream

of the rotor as a consequence of its displacement effect. Possibly the nacelle also contributes to this. This positive component decreases downstream in the wake as the expansion becomes asymptotic. All simulations, but the standard ACL, capture the general shape very well. In the latter a significant underprediction is visible already upstream of the rotor. The smaller induction results in a smaller blockage and thus a reduced deflection of the outer flow around the rotor.

The spikes in the distribution stem again from crossing the vortex sheet of the viscous wake. Based on the circulation

distribution, a positive amplitude can be expected from the former pressure side and a negative one from the suction side. Overall, these alternations are predicted by all simulations within standard deviation of the experiments.

### 5.5 The tip vortices

The development of the tip vortices will be mainly compared with the experiments of the first MEXICO campaign (Schepers and Snel, 2007). The reason for this is that only in the first campaign the PIV planes were placed such that they covered the tip

vortex development. [4]. In first MEXICO campaign, however, the entire span of the blade was equipped with boundary-layer tripping. Therefore, it should be noted that there is a small inconsistency to the simulations, in which the outer flow section

---

[4]The NewMexico experiments were only used to compare the first vortex referred to as $\boxed{\text{vortex 0}}$ in Fig. 17. It was extracted from the PIV sheet $20°$ azimuthally downstream of the blade (cf. Fig. 11)




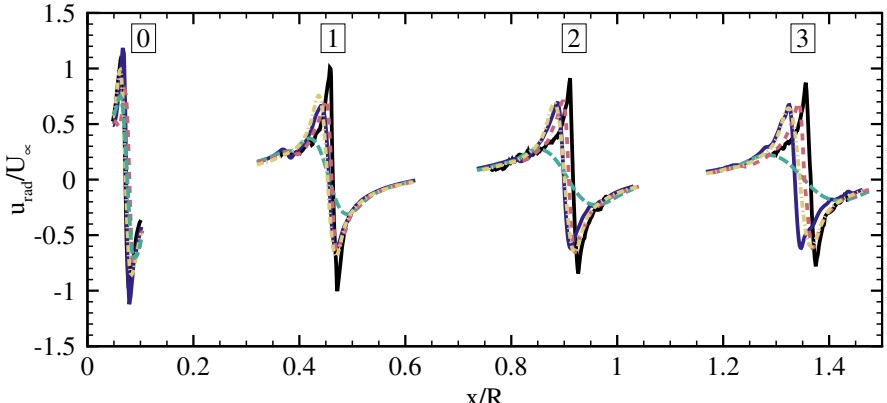

**Figure 17.** Comparison of the radial velocity in the tip vortex core. Experiment (Schepers and Snel, 2007) (——), Simulations: FR-BDES (——), FR-URANS (‑ ‑ ‑), ACL-forced (‧‧‧‧‧), ACL (‑ ‑ ‑).

was treated as transitional. However, due to the high angles of attack, the effect of the transition on the loads in the outer blade region is minor. Hence, no major differences in the circulation and thus on the tip vortex development are expected.

A first comparison of the vortex development shall be given based on the radial velocity (w.r.t. to the rotor, which is the
approximate circumferential direction of the tip vortex) traversing the vortex core in Fig. 17. The curves are extracted when the blade is nominally cutting the PIV plane, except for the reference $\boxed{\text{vortex 0}}$, which is evaluated at an azimuth offset of $20°$ (cf. Fig. 11). As with the radial traverses, the streamwise coordinate in the experiment is corrected by $-0.045\,m$ according to Schulz et al. (2016). With this correction, the streamwise position of the tip vortex agrees well for $\boxed{\text{vortex 0}}$ and $\boxed{\text{vortex 1}}$. Farther downstream, the FR-BDES simulation in especially lags slightly behind the experiments, suggesting a modestly slower
propagation velocity.

The velocity jump in $\boxed{\text{vortex 0}}$ is well captured by all simulations. Due to the higher grid resolution compared to the PIV sheet in that region the gradient is even larger in the scale-resolved simulations than in the experiments. Only, the FR-URANS simulation shows already a lower velocity jump. All simulations underpredict the amplitude of the experiments for $\boxed{\text{vortex 1}}$. Here, the grid spacing is around 6.6% of the tip chord (stays constant up to $\boxed{\text{vortex 3}}$) and obviously too coarse to accurately
capture the very sharp gradients in the vicinity of the vortex center. While the FR-BDES and the ACL-type simulations yield very similar maxima (with relatively higher values in the ACL-forced simulation), the FR-URANS simulation dramatically blurs the velocity across the vortex. This could already be expected from the previous examinations on the velocity field of the blade and rotor wakes. This trend prevails also for $\boxed{\text{vortex 2}}$ and $\boxed{\text{vortex 3}}$. It is interesting to note that the relative difference between all simulations (except the FR-URANS) and the experiment slightly decreases. By looking at the decay rates of the
velocity jumps in more detail (summarized in Tab. 3), some indication can be gained on whether it is the physical aging of the vortex or numerical dissipation that changes the amplitude. The decay rates of the semi-empirical model of van der Wall et al. (2016) are used for comparison as a further reference. In this model the aging of the vortex core with respect to the azimuth



**Table 3.** Comparison of vortex aging between experiments, simulations and the semi-empirical model of van der Wall et al. (2016).

| Case | Vortex 0 to 1 | Vortex 1 to 2 | Vortex 2 to 3 |
|------|---------------|---------------|---------------|
| Exp. | -2.7% | -12.8% | -6.4% |
| FR-BDES | -67.5% | +2.1% | -0.3% |
| FR-URANS | -111.8% | -24.7% | -17.1% |
| ACL-forced | -14.3% | -0.2% | -4.2% |
| ACL | -26.89% | -8.2% | -6.8% |
| semi-empirical | -2.61% | -2.48% | -2.36% |
| semi-empirical | -98.9% | -37.7% | -21.36%height |

angle $\psi$ behind the blade is defined as

$$r_{core} = r_{core,0}\sqrt{1 + \frac{5 \cdot 10^{-6}}{\left(r_{core,0}/R\right)^2 \Omega}\psi}, \tag{1}$$

where $r_{core,0}$ is the initial vortex core size assumed by van der Wall et al. (2016) as 5% of the tip reference at 93% rotor radius ($= 0.095\,m$) and $\Omega$ is the rotational speed of the rotor.

In the experiments, there is a very small reduction in the velocity jump (-2.7%) from ⬚vortex 0 to ⬚vortex 1 followed by a larger reduction (-12.8%) to ⬚vortex 2 and then again a smaller reduction (-6.4%) to ⬚vortex 3 . The small decay from ⬚vortex 0 to ⬚vortex 1 indicates that the PIV resolution is too coarse to fully resolve the vortex core. This was already sug-

gested by the PIV planes when we compared the blade wake in section 5.3. Furthermore, it is also expected from the semi-empirical model that the decay rates are largest at early stages and then decrease with propagation time in a square root relation. The decay rates of the van der Wall model are overall smaller than for the experiment. In all simulations the significant decay from ⬚vortex 0 to ⬚vortex 1 is due to the increased grid spacing and especially in case of the FR-URANS, additionally due to the diffusion introduced by the turbulence model. In both ACL simulations the decay rate is lower than in the FR-BDES

simulation since the initial vortex is larger due the smearing of the actuation forces. From ⬚vortex 1 to ⬚vortex 2 and beyond, the decay rates of the FR-BDES and the ACL-forced simulation are lower than in the experiment. It seems that vortex growth is suppressed as long as numerical dissipation and diffusion through the turbulence model are small, and vortex resolution is limited only by a grid too coarse to fully resolve the physical size.

In order to obtain a quantitative measure for the quality of the different simulation methods vortex position, circulation, core

radius and the core vorticity shall be evaluated with the tip vortex analysis tool developed at IAG (Meister et al., 2011). This tecplot tool uses velocity slices as input and can be easily used with different sources data, e.g. CFD, PIV or free vortex wake codes (Boorsma et al., 2018). The results are presented in Fig. 18. Two sets of measurement results are included. First, processing based on the phase-averaged velocity field, which was the standard procedure in the MexNext projects, and secondly, based on an instantaneous flow field. The simulations are evaluated from instantaneous fields. Besides of the simulations focused in

the present study, two further cases of the same authors are included which were conducted during the MexNext III project.





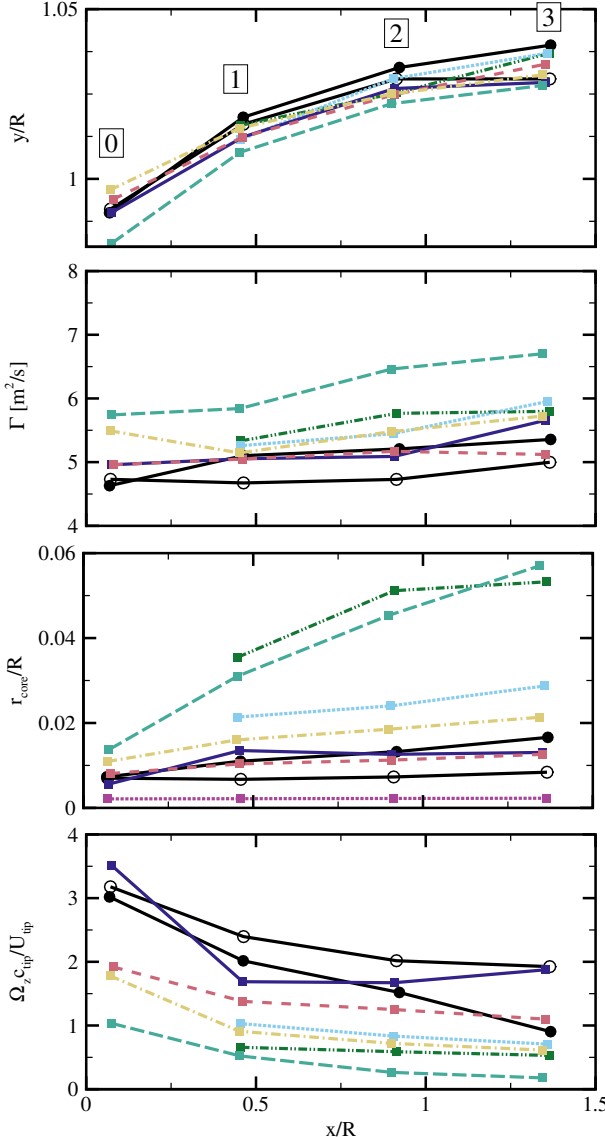

**Figure 18.** Comparison of the tip vortex properties between. Experiment (Schepers and Snel, 2007): time averaged (●), instantaneous (○). Simulations: FR-BDES (——), FR-URANS (— —), ACL-forced (— — —), ACL (— · — ·), FR-IDDES-WENO (MexNext III) (· · · · ·), FR-IDDES-JST (MexNext III) (— · · —), van der Wall et al. (2016) model (· · · · ·).

These are IDDES simulations conducted on a full turbine model, but on a wake grid being approximately twice as coarse as in the present study. One simulation was performed with a standard second order central difference scheme (JST) and one with a higher-order WENO scheme as in the present study. The results from these simulations help to understand the role of the turbulence modeling, grid resolution and numerical scheme on the preservation of the tip vortices.





The vortex core position depicted in the top graph of Fig. 18 is defined as the point of minimum out of plane vorticity. Note that the scale of the y-axis in the plot is 0.1 times the x-axis. The vortex core position is adequately captured by the simulations, with the tendency of slightly underpredicting the wake expansion.

The circulation is calculated by integrating the vorticity within a circular area around the vortex core. The integration is executed by increasing the radii, until convergence is reached. This is equivalent to the closed line integration of the velocity
vector according to the Stokes theorem and reads

$$\Gamma = \oint_C \boldsymbol{u} \cdot \boldsymbol{ds} = \int_A \boldsymbol{\omega} \cdot \boldsymbol{n} \, dA. \tag{2}$$

The magnitude of the circulation is shown in the second panel from top in Fig. 18. All simulations tend to overestimate the circulation which confirms the general trend that most models overpredicted the lift near the blade tip. The overprediction is significantly larger for the RANS compared to the DES simulations. Towards vortex 3 core, the uncertainty of the simula-
tions increases. For example the ACL-forced simulation showed very good agreement with its fully resolved counterpart for vortex 1 and vortex 2 predicts a decrease in the circulation, while the fully resolved simulation tends to result in an increase. The reason might be that in the fully resolved simulations and also in the PIV sheets vorticity stemming from the shear layer which wraps around the main tip vortex is included in the integration.

The vortex-core radius is crucial to assess the vortex preservation. Its accurate prediction is of special interest for example
when it comes to wake-flight-vehicle-interactions (Bühler et al., 2018; Tomaszewski et al., 2018; van der Wall et al., 2016; Cormier et al., 2020), or for investigations on the turbulent breakdown of the wake. The procedure in the evaluation tool of Meister (2015) follows the definition from Vollmers (2001), in which the vortex core is determined as the half distance between the peak values of the tangential velocity in a cross section through the vortex core. As expected from the previous analyses, the FR-URANS simulation strongly overestimates the vortex core radius by around a factor of four. The wrong slope of the
vortex growth emphasizes the excessive numerical diffusion.

In order to separate the effects of numerical dissipation and diffusion due to the turbulence model in DES-type simulations, we will include our results presented the MexNex III project. This was conducted on a coarser mesh with about the double grid spacing in the tip vortex area. Two IDDES simulations were performed, one with the second order JST scheme using a dissipation coefficient $k_4 = 128$ and one with the higher-order WENO scheme. In the simulation with the JST scheme even a
larger core radius is obtained than with the FR-URANS simulation for vortex 1 and vortex 2. However, in the later stages the gradient of the growth reduces, yielding an overall more concentrated tip vortex. The combination of DES with WENO clearly improves the results, since the numerical dissipation from the convection scheme is reduced. The vortex is now only double the size of the experiments and reveals a very similar growth rate as in the phase-averaged PIV data. This means that the numerical dissipation dominated the effect of the turbulence model and underlines that low-dissipation numerical schemes
are a prerequisite to fully exploit the potential of scale-resolving simulation techniques.

In the present FR-BDES and ACL-forced simulations the vortex diameter is significantly reduced. It is in the order of the phase-averaged PIV measurements, but still larger compared to the instantaneous PIV evaluation. The semi-empirical van der





Wall model predicts even smaller core radii than the instantaneous PIV results. This trend was also observed in Cormier et al. (2020), where it was compared to measurements and CFD simulations on a multi-megawatt turbine.

Turning finally to the maximum of the vorticity based on the maximum value found in the core region, a very similar trend holds as for the core radius. The FR-URANS solution severely underestimates the core vorticity. A clear improvement is obtained with DES as the eddy viscosity is effectively reduced. As observed before, the use of a higher-order scheme is beneficial to preserve the velocity gradients in the core. The FR-BDES simulation yields the most accurate results predicting even a larger vorticity than the experiments for the initial core of $\boxed{\text{vortex 0}}$. The smearing of the forces in the ACL-type

simulation already leads to a reduction of the vorticity. Due to the coarsening of the grid size from $\boxed{\text{vortex 0}}$ to $\boxed{\text{vortex 1}}$ the vorticity drops below the experimental values for all numerical approaches, but recovers for the FR-BDES until $\boxed{\text{vortex 3}}$. At that position, there is also a significant difference among the experimental results. The averaged data show an almost linear decrease of the vorticity, whereas in the instantaneous fields the slope decreases with the streamwise distance. The reason is that with the averaging, small scale vortex motions (vortex jittering or wandering) decrease their sharpness when looking at

them in a time-averaged frame.

Compared to numerical wake simulations found in literature, the FR-BDES simulation significantly improves prediction capabilities of tip vortices of wind-turbine wakes because it combines several methodological extensions, namely a higher-order numerical scheme in combination with the BDES turbulence modeling approach that results in almost implicit LES behavior in flows of solid rotation. In the results of Schulz et al. (2016), who used a three times coarser grid resolution in

the tip region than the present simulations together with URANS, the core radius was overestimated by a factor of five. In simulations of Nilsson et al. (2015), who employed the ACL method in an LES framework on a grid comparable to the IDDES simulations of MexNext III, it was overestimated by a factor of five as well. Meister (2015) employed a vortex refinement mesh together with the WENO scheme with URANS. By that, the overprediction could be reduced to a factor between three and four. In the studies of Carrión et al. (2014, 2015), a grid resolution in the wake comparable to the present one has been used. They

employed the low-Mach number adapted Roe-scheme with URANS and overpredicted the vortex core by a factor of around three. All refer to a comparison with phase-averaged PIV results. It can be summarized, also by own experiences, that with the URANS approach, successive grid refinement and the usage of low dissipation numerical schemes can reduce the vortex core diameter to a certain extent. However, particularly the slope of the vortex growth was always highly overpredicted which would result in inadequate predictions for example of wake breakdown. Accurate predictions of blade tip vortices therefore

require sufficient grid resolution and an adapted time step, as well as a turbulence model that does not erroneously damp the velocity gradients within the tip vortices. These methods further benefit from utilizing higher-order convection schemes. The fact, whether an actuator line method or geometrically fully resolved blade simulations are employed, plays only a subordinate role, as far as not the very early stages of the tip vortex development are of interest.





# 6    Conclusions

A computational study was conducted on the near wake and tip vortex development of the MEXICO model wind turbine operating in stalled conditions at low tip-speed ratio. Different simulation methodologies were compared with experimental data from PIV. One focus was on comparing the traditional URANS approach with a recently developed variant of the detached-eddy simulation technique. The second emphasis was on comparing the modeling of the rotor. In this, the actuator line method was juxtaposed with the simulation of the geometrically fully resolved turbine. Two approaches were pursued for the actuator

line simulation. The effects were isolated to the propagation of the injected forces by transferring them from the full-resolution simulation to eliminate the uncertainties of the BEM and the polars and comparing these results to a standard actuator line with internal force calculation.

The following conclusions can be drawn from this study:

1. The URANS turbulence modeling approach is sufficiently accurate to predict the mean velocity distribution in the near

wake governing the induction of the rotor. Expectedly, no interaction of turbulent structures in the blade wake was resolved. Also, the flow in region of the root vortex remained coherent. The tip vortex size was tremendously overpredicted, although sufficient grid resolution and a higher-order numerical scheme was provided. The URANS approach is though sufficiently accurate in predicting the mean induction effects of the blade wake. It can therefore still be recommended for practical applications focusing on the mean flow field of the near wake, even in the separated flow regime.

2. The scale resolving simulation using the Bernoulli-based DES formulation on the geometrically fully resolved rotor captured the details of the turbulent flow field in the vicinity of the blade, showing very good agreement regarding the shape and sizes of the structures. At the scale of the rotor wake, the axial velocity traverses were also found to coincide well with the experiments. The vortex core diameter agreed well with the phase-averaged PIV results, but is still overpredicted relative to the instantaneous fields. Moreover, secondary secondary vortices around the main tip vortex

could be captured. The formation of these structures as well as the dynamics of the tip vortex motion will be investigated in detail in part II of the paper.

3. The test of injecting the forces from blade resolved simulations was successful and could verify the method of the force projection. In the near-field of the actuator line, the same bound circulation could be proven as in the donor simulation. Accordingly, there was very good agreement between the two simulations with respect to the mean velocity field, the

tip vortex diameter and the mixing process of the root vortex in the inner region of the blade. However, since only the trailing vortices are resolved by the actuator line method, the development of the blade wake remained very concentrated and its breakdown was delayed.

4. In the standard actuator line approach with forces calculated from the blade-element momentum theory, the agreement of the loads in the outer blade area was reasonably accurate, but in the inner half of the rotor they were substantially

underestimated. This resulted in the formation of an incorrect wake topology in the inboard region, where the axial and tangential blade induction were predicted far too low. This could be remedied by using a model to account for the effect



rotational lift augmentation. In the outer region of the wake, the results were comparable to those of the other methods. However, the details of the trailing vortices in the middle region of the wake were not captured. Despite the deviations reported for the standard actuator line, it is an important conclusion that even in that complex flow situation with massive

separation the method predicts the mid and outer blade region reasonably accurate already at very short distances to the blade. This holds also for the prediction of the tip vortices which strength and size is captured in good match with the experiments. Typically, when applying the actuator line method in atmospheric turbulence the dynamic break-down of the wake is less delayed than in uniform inflow and closely agrees with fully resolved simulations. Therefore this method can be considered as basis to further develop engineering wake models for the near wake.

*Author contributions.* PW created the simulation setup of the fully resolved simulations, performed the simulations evaluated the simulations was the main contributor in writing the paper. MC created the ACL setup, performed and evaluated the simulations and wrote parts of the paper related to the ACL. TL and EK supervised the work and revised the manuscript.

*Competing interests.* The authors declare that they have no conflict of interest.

*Acknowledgements.* The authors gratefully acknowledge the vital discussions on aerodynamics within the consortium of the former IEA Task
29 and the High-Performance Computing Center Stuttgart (HLRS) for providing computational resources within the project WEALoads.





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
