# Peer review of "The near wake development of a wind turbine operating in stalled conditions - Part I: Assessment of numerical models"

_Wind Energy Science, 2023_

## Author Response (AR1)

**Reply to comments by Reviewer 1**

**Pascal Weihing**

**10. Januar 2024**

The authors would like to thank Dr. Mockett for his efforts and valuable comments. They are very much appreciated and incorporated into the revised paper. In the present document the comments given by Dr. Mockett are addressed consecutively. The following formatting is chosen:

- The reviewer comments are marked in blue and italic.

- The reply by the author is in black color

- A marked-up manuscript is added. Changed sections with regard to the comments by reviewer 1 are marked in yellow. Changed sections with regard to comments by both reviewers are marked in green. Highlighting in gray denotes passages that have been changed by the author in order to improve the clarity or the argumentation but which are not related to specific reviewer comments.

**Specific comments**

1. "*Lines 220-225: The localised Bernoulli equation in the BDES is not Galilean invariant. Please add a few lines explaining how this is overcome for the moving geometry.*"

Thank you for pointing out this aspect which is particularly relevant for the rotary wing setup. An explanation was added at $\boxed{\textbf{R1:1.1}}$ (page 9, line 224).

2. "*Lines 315-318: I haven't understood the boundary conditions on the front and rear part of the nacelle. Is this just a rotating wall velocity BC, or something else? Please adjust for clarity.*"

We were not quite sure whether you are referring to the small patch which was introduced to avoid the singular line, or the brown faces that shall be treated as non-rotating walls. Regarding the first one we used a simple slip-wall condition. It rotates as the grid is moving, but does not impose any velocities at te boundary. Concerning the second point we tried to make the description a bit clearer. Please see $\boxed{\textbf{R1:2.1}}$ (page 12, line 322)

3. "*In the sections comparing simulation and PIV, key information is buried too late in the text. Please place explanations of the following more prominently: Are the results instantaneous or phase-averaged in the CFD and PIV? How do the locations in degrees relate to the position of the blade (e.g. are the snapshots always taken when the blade is at "top dead centre"?). Information on whether data is instantaneous or phase-averaged should be added to the relevant figure captions.*"

Further general information on the PIV sheets was added in the section describing the test case (see $\boxed{\textbf{R1:3.1}}$ (page 6, line 171)). The information on the PIV phase angles relative the the blade, the locations of the PIV sheets and whether data are based on instantaneous or averaged fields were described in the beginning of each section discussing the PIV and CFD results:

- Comparison of the flow structures downstream of the blade: **R1:3.2** (page 22, line 519)
- Comparison of the radial traverses: **R1:3.3** (page 28, line 619) and **R1:3.4** (page 28, line 627)
- Comparison of the axial traverses: **R1:3.5** (page 31, line 695)
- Comparison of the tip vortices: **R1:3.6** (page 35, line 764)

The captions of Fig. 11-18 were adapted including the information on whether instantaneous or phase-averaged data were used.

Additionally, small sketches were included in the Figs. 11-16, in order to visualize the relative position of the blade of interest w.r.t. to the PIV plane.

4. "*Figure 17 and discussion: Please explain whether the radial position of the profiles is adjusted to the actual vortex location in each snapshot, or at a fixed location. This is important, as you observe later, since the vortex location wanders from cycle to cycle.*"

An explanation on the radial extraction position was added at **R1:4.1** (page 35, line 768). It was indeed at the position of maximum vorticity for each of the cases, respectively. Please note that the focus of this evaluation was on the change of the velocity amplitude. Therefore, instantaneous data was used rather than the phase-averaged fields. Vortex jittering effects were neglected, however, they may lead to slight changes in the x-position of the vortex core. A statement on this effect is added at **R1:4.2** (page 35, line 774). Concerning the vortex jittering, a more detailed analysis will be shown in part II of the study.

5. "*Lines 848-849: Please expand slightly. What aspect of BDES causes "almost implicit LES behaviour in flows of solid rotation"? The sigma-model form of the SGS mode?*"

As you have pointed out, it is the $\sigma$-velocity gradient operator utilized in the DES model that does not introduce eddy viscosity in flows with pure rotation. A statement on that was added in the revised manuscript at **R1:5.1** (page 39, line 867).

**Technical corrections**

We would also like to thank you very much for the editorial corrections. We have corrected them all according to your suggestions. We have also proofread the entire manuscript for orthography.

1. "*The word "though" is often used incorrectly. "However" or "nonetheless" are suggested instead, e.g. Lines 369, 596, 678, 877*" -> Corrected

2. "*Line 508: origin -> originate*" -> Corrected

3. "*Line 608: "pretend" isn't the right word, suggest "occur"*" -> Corrected

4. "*Line 630: on behalf of -> for*" -> Corrected

5. "*Line 682: matter -> consequence*" -> Corrected

6. "*Line 707: but -> except*" -> Corrected

7. "*Lines 726-727: "which results that" is unclear, please revisit this sentence*"

The sentence was reformulated to: The induction factor is only $a_{ax} = 0.13$ at $r/R = 0.8$. This means that the average velocity is still about $75\,\%$ of the inflow velocity **R1:7.1** (page 34, line 738).

**Reply to comments by Reviewer**

Pascal Weihing

January 10, 2024

The authors would like to thank the second reviewer for her/his efforts and valuable comments. They are very much appreciated and incorporated into the revised paper. In the present document the comments given by the second reviewer are addressed consecutively. The following formatting is chosen:

- The reviewer comments are marked in blue and italic.

- The reply by the author is in black color

- A marked-up manuscript is added. Changed sections with regard to the comments by reviewer 2 are marked in blue. Changed sections with regard to comments by both reviewers are marked in green. Highlighting in gray denotes passages that have been changed by the author in order to improve the clarity or the argumentation but which are not related to specific reviewer comments.

**Specific comments**

We are aware that the manuscript is long. We have done our best to maintain a reasonably clear structure. In the revised manuscript the paragraph with the objectives and the conclusions have been sharpened.

1. "*I suggest removing all the results with the standard ACL (ACL-polars) method or redoing the simulations with appropriate airfoil data. It does not make much sense to make a comparison with uncorrected 2D airfoil data that you already know will not work well for this case*"

We understand that removing all the results with the standard ACL method would certainly shorten the manuscript. Also your comment is valid that one could have assumed before that without correction models the ACL with standard polars will lead to inacurracies in the wake.

For the following reasons, we hesitate to remove the results from the manuscript. However, we would still do so, should you otherwise reject the paper.

The reason for using the 2D measured polars as input for the ACL was because it was agreed within MexNext III that all partners should use the same "official" polar set without any corrections for the validation rounds. A comment on the origin of the data was made in $\boxed{\textbf{R2:1.2}}$ (page 12, line 310).

Although also we had assumed before that the use of the uncorrected data will lead to inaccuracies in the wake prediction, we still consider it important to quantify these effects and to trace the reasons for the observed behavior. To the best of our knowledge, the effect of force determination in the ACL model (i.e., whether based on uncorrected 2D polars vs. "correct" forces from higher fidelity simulation) on wake behavior has not yet been studied in detail. Therefore, we consider it valuable to show that with uncorrected 2D polars there is a significant change in

the flow topology of the wake, while imposing the correct forces results in nearly the same wake behavior as in the fully resolved parent simulations, also in massively separated flow which was not clear before. The important conclusion is that we can apply the ACL to stalled scenarios, but the forces must be as close as possible to the actual 3D case. Removing the results with the 2D polars would leave the ACL results with the specified forces pretty much alone without context.

It would be a good addition to also include the result with the corrected polars as an example case. However, this was not possible in the short time available due to the very high computational requirements. It could be the subject of future work.

2. "*In the abstract you write that "...  URANS method performed very poorly". This is not covering the actual shown performance of the URANS and is inconsistent with what you later write in the summary (5.3.3) and the conclusions.*"

Thank you for this comment. Perhaps this sentence could have been easily be misinterpreted. The word "poorly" referred only to the prediction quality of the tip vortex properties. The satisfactory quality of the URANS method in predicting the wake deficit was meant to be expressed by the statement in line 12 of the original manuscript, where we had written that the general quantities of the near wake are well predicted by all methods. We have now made the statements more precise to avoid misinterpretation, by summarizing for each simulation method which phenomena can be described well and which less well $\boxed{\textbf{R2:2.1}}$ (page 1, line 12).

3. "*Line 615: What do you mean by phase averaging? Your only simulate one rotor revolution. Is it the equivalent of averaging over three blades?*"

Our "definition" on the phase-averaging was expanded briefly in $\boxed{\textbf{R2:3.1}}$ (page 28, line 627) of the revised manuscript. Here, we would like to give are more detailed answer on our data extraction and averaging procedure. We started to extract the flow solution over one revolution, after completion of the initial transient, which included a steady-state pre-simulation and several revolutions at larger time step sizes. Time averaging was conducted over this one revolution. The velocity was averaged in the moving frame of reference of the blade. Data were then extracted with respect to the phase angles relative to the blade. As long as the flow field is rotationally symmetric, which is the case in the present simulation, this is equivalent to ensemble averaging in the inertial system over several revolutions at a specific phase angle relative to the blade. However, it should be noted that same results are expected for sufficiently long averaging time. This averaging time must be large enough to achieve a converged mean value also for the largest turbulence scales in the blade wake. We consider one revolution of averaging time sufficiently long as it represents about 140 flow passes in the outer blade region and about flow passes in the inner part of the rotor.

**Editorial comments**

1. "*In general the English writing is good but there are still many sentences in the manuscript with missing words and there are cases with duplicate words and (nearly) duplicate sentences (e.g. line 80-85 and 850-855). I suggest a careful read through to correct them.*"

Thank you for the hints regarding the duplicate sentences and that we still have missing words in many sentences. The former lines 850-855 have been removed with referring to the introduction $\boxed{\textbf{R2:4.1}}$ (page 39, line 864). We have also proofread the entire manuscript for orthography.